# PALM: Preference-based Adversarial Manipulation against Deep Reinforcement Learning

## Abstract

Deep reinforcement learning (DRL) methods are vulnerable to adversarial attacks such as perturbing observations with imperceptible noises. To improve the robustness of DRL agents, it is important to study their vulnerability under adversarial attacks that would lead to extreme behaviors desired by adversaries. Preference-based RL (PbRL) aims for learning desired behaviors with human preferences. In this paper, we propose **PALM**, a preference-based adversarial manipulation method against DRL agents which adopts human preferences to perform targeted attacks with the assistance of an intention policy and a weighting function. The intention policy is trained based on the PbRL framework to guide the adversarial policy to mitigate restrictions of the victim policy during exploration, and the weighting function learns weight assignment to improve the performance of the adversarial policy. Theoretical analysis demonstrates that PALM converges to critical points under some mild conditions. Empirical results on a few manipulation tasks of Meta-world show that PALM exceeds the performance of state-of-the-art adversarial attack methods under the targeted setting. Additionally, we show the vulnerability of the offline RL agents by fooling them into behaving as human desires on several Mujoco tasks. Our code and videos are available in `https://sites.google.com/view/palm-adversarial-attack`.

## 1 Introduction

Adversarial examples in image classifiers have prompted a new field of studying the vulnerability of deep neural networks (DNN). Recent researches demonstrate that reinforcement learning (RL) agents parameterized by DNN also show vulnerability under adversarial attacks (Huang et al., 2017; Pattanaik et al., 2018; Zhang et al., 2020; 2021; Sun et al., 2022). Adversaries generate imperceptible perturbations to the observations of victim agent, making agents fail to complete the original behaviors. While adversarial attack is an crucial approach to evaluate the vulnerability of the agents, targeted attack in RL has received little attention. Recently, embodied intelligence (Gupta et al., 2021; Liu et al., 2022; Ahn et al., 2022; Reed et al., 2022; Fan et al., 2022) is considered as a meaningful way to improve the cognitive ability of artificial intelligence. These embodied agents show powerful capabilities while possibly exposing vulnerability. Therefore, we wonder that:

**How can one manipulate the agent to perform desired behaviors, and whether the embodied agents are robust to adversarial manipulations?**

To achieve targeted adversarial attacks, one straight way is to design respective rewards for the adversary agents. However, specifying a precise reward function can be challenging. For example, it is difficult to design a reward function denoting the goodness of the current step in the game of Go. For example, it is difficult in chess games to craft a reward function which can identify the quality of each move. In preference-based RL framework, a human only needs to provide binary preference labels over two trajectories of the agent (Christiano et al., 2017). Compared to reward engineering, preference-based RL is an easier way to learn policies through human preferences. Meanwhile, recent research on preference-based RL shows an excellent capacity to learn novel behaviors with few preference labels (Lee et al., 2021a; Park et al., 2022; Liang et al., 2022), and significantly improves feedback efficiency.

Motivated by this, we consider using preference-based RL to perform targeted adversarial attacks from the following perspectives. On the one hand, it is difficult to define the desired behaviors to achieve targeted attacks, but humans can implicitly inject intentions by providing binary preferences.

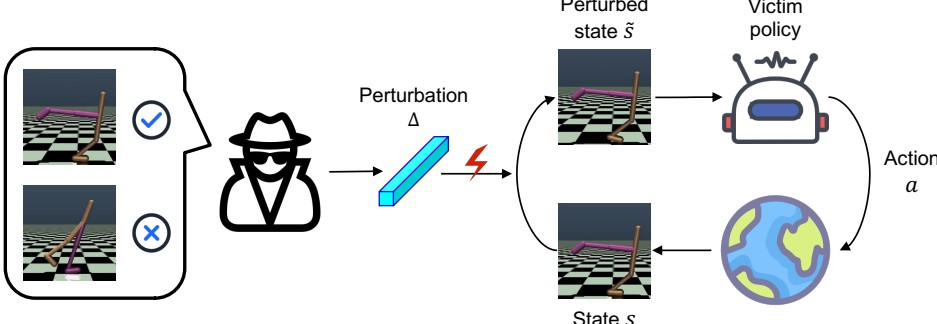

Figure 1: Illustration of targeted attack from PALM. The adversary first receives the true state s from the environment and perturbs it into s̃. Then the victim observes s̃ and takes action according to it.

On the other hand, preference-based RL is a data-efficient manipulation method because a few preference labels are enough to learn well-behaved policies. As shown in Figure 1, we emphasize PALM can recover the reward function of desired behaviors via preference-based RL framework and manipulate agents to perform human desired behaviors.

In this paper, we propose **P**reference-based **A**dversaria**L** **M**anipulation (PALM) algorithm, which performs targeted attack from human preferences. PALM includes an adversary to perturb the observations of the victim. To achieve targeted attack and better exploration, we introduce an intention policy, which is learned from human preferences, as the learning target of the adversary. Additionally, we utilize a weighting function to assist adversary learning by re-weighting state examples.

In summary, our contributions are three-fold. Firstly, to the best of our knowledge, we propose the first targeted adversarial attack method against DRL agents via preference-based RL. Secondly, we theoretically analyze PALM and provide a convergence guarantee under some mild conditions. Lastly, we design two scenarios and experiments on Meta-world demonstrate that PALM outperforms the baselines by a large margin. Empirical results demonstrate that both online and offline RL agents are vulnerable to our proposed adversarial attacks.

## 2  RELATED WORK

Many previous works on adversarial attacks study the vulnerability of a DRL agent. Huang et al. (2017) computes adversarial perturbations via utilizing the technique of FGSM (Goodfellow et al., 2015) to mislead the victim policy, not to choose the optimal action. Pattanaik et al. (2018) presents an approach that leads the victim to select the worst action based on the Q-function of the victim. Gleave et al. (2020) conducts adversarial attacks under the two-player Markov game instead of perturbing the agent's observation. Zhang et al. (2020) proposes the state-adversarial MDP (SA-MDP) and develops two adversarial attack methods named Robust Sarsa (RS) and Maximal Action Difference (MAD). SA-RL (Zhang et al., 2021) directly optimizes the adversary policy to perturb state in the form of end-to-end RL. PA-AD (Sun et al., 2022) designs an RL-based "director" to find the optimal policy perturbing direction and construct an optimized-based "actor" to craft perturbed states according to the given direction. Methods of untargeted adversarial attack focus on making the victim policy fail, while our approach emphasizes manipulating the victim policy. That is to say, victim's behaviors are consistent with the preference of the manipulator under attacks. Another line of works (Pinto et al., 2017; Mandlekar et al., 2017; Pattanaik et al., 2018) consider using adversarial examples to improve the robustness of policies, although it is out of the scope of this paper.

There are a few prior researches that focus on targeted attacks on RL agents. Lin et al. (2017) first proposes a targeted adversarial attack method against DRL agents. It attacks the agent to reach a targeted state. Buddareddygari et al. (2022) also present a strategy to mislead the agent towards to a specific state by placing an object in the environment. The hijacking attack (Boloor et al., 2020) is proposed to attack agents to perform targeted actions on autonomous driving systems. Hussenot et al. (2019) provides a new perspective that attacks the agent to imitate a target policy. Our method differs that PALM manipulates victim behave as human desire and focuses on the preference-based RL. Xiao et al. (2019) proposes the first adversarial attack method against real world visual navigation robot. Lee et al. (2021b) investigates targeted adversarial attacks against the action space of the agent. Our method differs that PALM leverages preference-based RL to avoid reward engineering

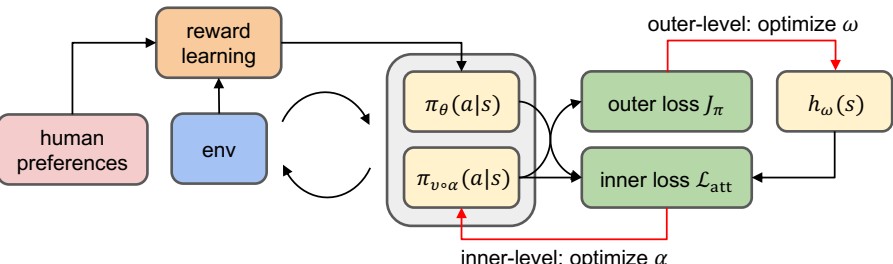

Figure 2: Framework of PALM. PALM jointly learns an intention policy, an adversary and a weighting function under bi-level optimization framework. In the inner-level, the adversary is optimized to approach the intention policy which learns via preference-based RL. In the outer-level, the weighting function is updated to maximize the performance of the adversary evaluated by the outer loss.

and learns an intention policy to tackle restricted exploration problem, so that PALM can attack the victim policy to perform behaviors far from its original behaviors.

Training agents with human feedback has been investigated in several works. Preference-based RL provides an effective way to utilize human preferences for agent learning. Christiano et al. (2017) proposes a basic learning framework for preference-based RL. To further improve feedback efficiency, Ibarz et al. (2018) additionally utilizes expert demonstrations to initialize the policy besides learning the reward model from human preferences. However, previous methods need plenty of human feedback, which is usually impractical. Many recent works have proposed to tackle this problem. Lee et al. (2021a) presents a feedback-efficient preference-based RL algorithm, which benefits from unsupervised exploration and reward relabeling. Park et al. (2022) further improves feedback efficiency by semi-supervised reward learning and data augmentation, while Liang et al. (2022) proposes an intrinsic reward to enhance exploration. To the best of our knowledge, our method is the first to achieve a targeted adversarial attack against DRL agents through preference-based RL.

## 3 PROBLEM SETUP

**The Victim Policy**. In RL, agent learning can be modeled as a finite horizon Markov Decision Process (MDP) defined as a tuple $(\mathcal{S}, \mathcal{A}, \mathcal{R}, \mathcal{P}, \gamma)$. $\mathcal{S}$ and $\mathcal{A}$ denote state and action space, respectively. $\mathcal{R} : \mathcal{S} \times \mathcal{A} \times \mathcal{S} \rightarrow \mathbb{R}$ is the reward function and $\gamma \in (0, 1)$ is the discount factor. $\mathcal{P} : \mathcal{S} \times \mathcal{A} \times \mathcal{S} \rightarrow [0, 1]$ denotes the transition dynamics, which determines the probability of transferring to $\mathbf{s}'$ given state $\mathbf{s}$ and action $\mathbf{a}$. We denote the stationary policy $\pi_\nu : \mathcal{S} \rightarrow \mathcal{P}(\mathcal{A})$, where $\nu$ are parameters of the victim. We suppose the victim policy is fixed and uses the approximator.

**The Adversarial Policy.** To study the adversary learning with human preferences, we formulate it as rewarded state-adversarial Markov Decision Process (RSA-MDP). Formally, a RSA-MDP is a tuple $(\mathcal{S}, \mathcal{A}, \mathcal{B}, \widehat{\mathcal{R}}, \mathcal{P}, \gamma)$. The adversary $\pi_\alpha : \mathcal{S} \rightarrow \mathcal{P}(\mathcal{S})$ perturbs the states before the victim observes them, where $\alpha$ are parameters of the adversary. Specifically, the adversary perturbs the state $\mathbf{s}$ into $\tilde{\mathbf{s}}$ which is restricted by $\mathcal{B}(\mathbf{s})$ (i.e., $\tilde{\mathbf{s}} \in \mathcal{B}(\mathbf{s})$). $\mathcal{B}(\mathbf{s})$ is defined as a small set $\{\tilde{\mathbf{s}} \in \mathcal{S} : \| \mathbf{s} - \tilde{\mathbf{s}} \|_p \leq \epsilon\}$, which limits the attack power of the adversary and $\epsilon$ is attack budget. Since directly generating $\tilde{\mathbf{s}} \in \mathcal{B}(\mathbf{s})$ is hard, the adversary learns to produce a Gaussian noise $\Delta$ with $\ell_\infty(\Delta)$ less than 1, and we obtain the perturbed state through $\tilde{\mathbf{s}} = \mathbf{s} + \Delta * \epsilon$. The victim takes action according to the observed $\tilde{\mathbf{s}}$, while true states in the environment are not changed. The perturbed policy is denoted as $\pi_{\nu \circ \alpha}$. Different from SA-MDP (Zhang et al., 2020), RSA-MDP introduces $\widehat{\mathcal{R}}$, which is consistent with human preferences. The target of RSA-MDP is to solve the optimal adversary $\pi_\alpha^*$, which enables the victim to achieve the maximum expected return over all states. Lemma 1 shows that solving the optimal adversary in RSA-MDP is equivalent to finding the optimal policy in MDP $\widehat{\mathcal{M}} = (\mathcal{S}, \hat{\mathcal{A}}, \widehat{\mathcal{R}}, \widehat{\mathcal{P}}, \gamma)$, where $\hat{\mathcal{A}} = \mathcal{S}$ and $\widehat{\mathcal{P}}$ is the transition dynamics of the adversary.

## 4 METHOD

In this section, we introduce our method PALM, which leverages preference-based RL to achieve targeted attack against DRL agents. The core idea of PALM, on the one hand, is to learn an intention policy as the learning target of the adversarial policy to tackle restricted exploration problem. On the other hand, PALM takes advantage of feedback-efficient preference-based RL method PEB-

BLE (Lee et al., 2021a) to avoid reward engineering. Also, we introduce a weighting function to improve the performance of the adversary and formulate PALM as a bi-level optimization algorithm. The framework of PALM is shown in Figure 2 and detailed procedure is summarized in Appendix A.

## 4.1 LEARNING INTENTION POLICY

PALM aims to find the optimal adversary that manipulates the victim's behaviors to be consistent with human intentions. However, the victim policy is pre-trained to complete a specific task, directly learning an adversary suffer from exploration difficulty caused by the restriction of victim policy, making it hard to find an expected adversarial policy efficiently. Therefore, we introduce an intention policy $\pi_\theta$ which has unrestricted exploration space to guide adversarial policy training.

To achieve targeted attack and avoid reward engineering, we inject human intentions into the intention policy via preference-based RL framework, which is shown in Figure 3. In preference-based RL, the agent have no access to the ground-truth reward function. To learn a reward function, humans provide preference labels between two trajectories of the agent and the reward function $\widehat{r}_\psi$ learns to align with the preferences (Christiano et al., 2017).

Formally, a segment $\sigma$ of length $k$ is denoted as a sequence of states and actions $\{\mathbf{s}_{t+1}, \mathbf{a}_{t+1}, \cdots, \mathbf{s}_{t+k}, \mathbf{a}_{t+k}\}$. A human expert is required to give a label $y$ of a pair of segments $(\sigma^0, \sigma^1)$ to indicate which segment is preferred, where $y \in \{(0,1), (1,0), (0.5, 0.5)\}$. Following

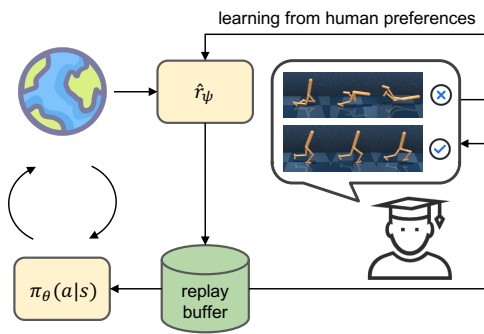

Figure 3: Diagram of preference-based RL.

Bradley-Terry model (Bradley & Terry, 1952), a preference predictor is constructed in (1):

$$P_\psi[\sigma^0 \succ \sigma^1] = \frac{\exp \sum_t \widehat{r}_\psi(\mathbf{s}_t^0, \mathbf{a}_t^0)}{\exp \sum_t \widehat{r}_\psi(\mathbf{s}_t^0, \mathbf{a}_t^0) + \exp \sum_t \widehat{r}_\psi(\mathbf{s}_t^1, \mathbf{a}_t^1)}, \tag{1}$$

where $\sigma^0 \succ \sigma^1$ denotes $\sigma^0$ is preferred to $\sigma^1$. This predictor indicates the probability that a segment is preferred is proportional to its exponential return. Then, the reward function is optimized by aligning the predicted preference labels with human preferences through cross-entropy loss:

$$\mathcal{L}(\psi) = - \mathop{\mathbb{E}}_{(\sigma^0, \sigma^1, y) \sim \mathcal{D}} \Big[ y(0) \log P_\psi[\sigma^0 \succ \sigma^1] + y(1) \log P_\psi[\sigma^1 \succ \sigma^0] \Big], \tag{2}$$

where $\mathcal{D}$ is a dataset of triplets $(\sigma^0, \sigma^1, y)$ consisting of segment pairs and human preference labels. By minimizing (2), we obtain a reward function estimator $\widehat{r}_\psi$, which is used to provide estimated rewards for agent learning via any RL algorithms. Following PEBBLE (Lee et al., 2021a), we use an off-policy actor-critic method SAC (Haarnoja et al., 2018) to learn a well-performing policy. Specifically, the Q-function $Q_\phi$ is optimized by minimizing the Bellman residual:

$$J_Q(\phi) = \mathop{\mathbb{E}}_{\tau_t \sim \mathcal{B}} \left[ \left( Q_\phi(\mathbf{s}_t, \mathbf{a}_t) - \widehat{r}_t - \gamma \bar{V}(\mathbf{s}_{t+1}) \right)^2 \right], \tag{3}$$

where $\bar{V}(\mathbf{s}_t) = \mathbb{E}_{\mathbf{a}_t \sim \pi_\theta} \left[ Q_{\bar{\phi}}(\mathbf{s}_t, \mathbf{a}_t) - \mu \log \pi_\theta(\mathbf{a}_t | \mathbf{s}_t) \right]$, $\tau_t = (\mathbf{s}_t, \mathbf{a}_t, \widehat{r}_t, \mathbf{s}_{t+1})$ is the transition at time step $t$, $\bar{\phi}$ is the parameter of the target soft Q-function. The policy $\pi_\theta$ is updated by minimizing (4):

$$J_\pi(\theta) = \mathbb{E}_{\mathbf{s}_t \sim \mathcal{B}, \mathbf{a}_t \sim \pi_\theta} \left[ \mu \log \pi_\theta(\mathbf{a}_t | \mathbf{s}_t) - Q_\phi(\mathbf{s}_t, \mathbf{a}_t) \right], \tag{4}$$

where $\mu$ is the temperature parameter. By learning an intention policy, PALM tackles restricted exploration problem and provides an attack target for the following adversary training.

## 4.2 LEARNING ADVERSARIAL POLICY AND WEIGHTING FUNCTION

To make the victim policy perform human desired behaviors, PALM learns the adversary by minimizing the KL divergence between the perturbed policy $\pi_{\nu \circ \alpha}$ and the intention policy $\pi_\theta$. However, different states may have various importance to induce the victim policy to the target. To stabilize training process and improve the performance of the adversary, we introduce a weighting function $h_\omega$ to re-weight states in adversary training.

We formulate PALM as a bi-level optimization algorithm, which alternately updates the adversarial policy $\pi_\alpha$ and the weighting function $h_\omega$ through inner and outer optimization. In the inner level, PALM optimizes $\alpha$ with the importance weights outputted by a weighting function $h_\omega$, and optimizes $\omega$ in the outer level according to the performance of the adversary. Intuitively, the adversary learns to approach the intention policy in the inner level, while the weighting function learns to improve the performance of the adversary by evaluating the performance of the adversary through a meta-level loss. The whole objective of PALM is:

$$
\begin{aligned}
\min_\omega \quad & J_\pi(\alpha(\omega)), \\
\text{s.t.} \quad & \alpha(\omega) = \arg\min_\alpha \mathcal{L}_{\text{att}}(\alpha; \omega, \theta).
\end{aligned}
\tag{5}
$$

**Inner-level Optimization: Training adversarial policy $\pi_\alpha$.** In the inner-level optimization, given the intention policy $\pi_\theta$ and the weighting function $h_\omega$, we hope to find the optimal adversarial policy by minimizing the re-weighted KL divergence between $\pi_{\nu\circ\alpha}$ and $\pi_\theta$ in (6):

$$
\mathcal{L}_{\text{att}}(\alpha; \omega, \theta) = \mathbb{E}_{\mathbf{s}\sim\mathcal{B}}\Big[ h_\omega(\mathbf{s}) D_{\text{KL}}\left( \pi_{\nu\circ\alpha}(\mathbf{s}) \parallel \pi_\theta(\mathbf{s}) \right) \Big],
\tag{6}
$$

where $h_\omega(\mathbf{s})$ is the importance weights outputted by the weighting function $h_\omega$. Intuitively, the adversarial policy is optimized to make the perturbed policy be close to the intention policy, while $h_\omega$ assigns different weights to states of various importance. With the collaborative assistance of the intention policy and the weighting function, PALM efficiently learns an optimal adversarial policy.

**Outer-level Optimization: Training weighting function $h_\omega$.** As for the outer-level optimization, we need to find a precise weighting function to balance the state distribution and assign proper weights to propel adversary learning. The weighting function is trained to distinguish the importance of states by evaluating the performance of the perturbed policy. Specifically, the perturbed policy $\pi_{\nu\circ\alpha}$ is evaluated using a policy loss in (7), which is adapted from the policy loss in (4):

$$
J_\pi(\alpha(\omega)) = \mathbb{E}_{\mathbf{s}_t\sim\mathcal{B}, \mathbf{a}_t\sim\pi_{\nu\circ\alpha(\omega)}} \left[ \mu \log \pi_{\nu\circ\alpha(\omega)}(\mathbf{a}_t|\mathbf{s}_t) - Q_\phi(\mathbf{s}_t, \mathbf{a}_t) \right],
\tag{7}
$$

where $\alpha(\omega)$ denotes $\alpha$ implicitly depends on $\omega$. Therefore, PALM calculates the implicit derivative of $J_\pi(\alpha(\omega))$ with respect to $\omega$ and finds the optimal $\omega^*$ by optimizing (7). To make it feasible, we make an approximation of $\arg\min_\alpha$ with the one-step gradient update. (8) obtains an estimated $\arg\min_\alpha$ with one-step updating and builds a connection between $\alpha$ and $\omega$:

$$
\hat{\alpha}(\omega) \approx \alpha_t - \eta_t \left. \nabla_\alpha \mathcal{L}_{\text{att}}(\alpha; \omega, \theta) \right|_{\alpha_t}.
\tag{8}
$$

According to the chain rule, the gradient of the outer loss with respect to $\omega$ can be expressed as:

$$
\left. \nabla_\omega J_\pi(\alpha(\omega)) \right|_{\omega_t} = \left. \nabla_{\hat{\alpha}} J_\pi(\hat{\alpha}(\omega)) \right|_{\hat{\alpha}_t} \left. \nabla_\omega \hat{\alpha}_t(\omega) \right|_{\omega_t} = \sum_{\mathbf{s}} f(\mathbf{s}) \cdot \left. \nabla_\omega h(\mathbf{s}) \right|_{\omega_t},
\tag{9}
$$

where $f(\mathbf{s}) = -\eta_t \cdot \left( \nabla_{\hat{\alpha}} J_\pi(\alpha(\omega)) \right)^\top \nabla_\alpha D_{\text{KL}}\left( \pi_{\nu\circ\alpha}(\mathbf{s}) \parallel \pi_\theta(\mathbf{s}) \right)$ and detailed derivation can be found in Appendix B. The key to obtain this meta gradient is building and computing the relationship between $\alpha$ and $\omega$. Obtaining the implicit derivative, PALM updates the parameters of the weighting function by taking gradient descent with outer learning rate.

In addition, we theoretically analyze the convergence of PALM in Theorem 1 and 2. In Theorem 1, we demonstrate the convergence rate of the outer loss, i.e. the gradient of the outer loss with respect to $\omega$ will convergence to zero. Thus PALM learns a more powerful adversary using importance weights outputted by the optimal weighting function. In Theorem 2, we prove the convergence of the inner loss. The inner loss of PALM algorithm converges to critical points under some mild conditions, which ensures the parameters of the adversary can converge to the optimal parameters. Theorems and proofs can be found in Appendix D.

## 5 EXPERIMENTS

In this section, we evaluate our method on several robotic simulated manipulation tasks from Metaworld (Yu et al., 2020) and continuous locomotion tasks from Mujoco (Todorov et al., 2012). Specifically, our experiment contains two essential phases. In the first phase, we verify the efficacy of the proposed method through two scenarios: navigation and opposite behaviors. Furthermore, we show the capability of our approach by fooling a popular offline RL method, Decision Transformer (Chen et al., 2021), into acting specific behaviors in the second phase. The detailed description of tasks used in experiments is provided in Appendix F.

## 5.1 SETUP

**Baselines.** We include the existing evasion attack methods for comparison to study the effectiveness of our approach.

- Random attack: this is a naive baseline that samples random perturbed observations via a uniform distribution.

- SA-RL (Zhang et al., 2021): this method learns an adversarial policy in the form of an end-to-end RL formulation.

- PA-AD (Sun et al., 2022): this method combines RL-based "director" and non-RL "actor" to find state perturbations, which is the state-of-the-art adversarial attack algorithm against DRL.

- PALM: our proposed method, which collaboratively learns adversarial policy and weighting function with the guidance of intention policy.

**Implementation Settings.** We compare PALM with existing adversarial attack methods, which attack the victim to reduce its cumulative reward rather than manipulate it. To achieve fair comparison, we make simple adjustments for SA-RL and PA-AD to suit our settings in the experiments. In their original version, both of these two methods use the negative value of the reward obtained by the victim to train an adversary. We replace it with the same estimated reward function $\widehat{r}_\psi$ as our method uses, which means they learn from human preferences. Following the settings in PEBBLE (Lee et al., 2021a), we use a scripted teacher that provides ground truth preference labels. More details of scripted teacher and preference collection can be found in Appendix E. For the implementation of SA-RL[1] and PA-AD[2], we use the released official codebase. For fair comparison, all methods learned via preference-based RL are given the same number of preference labels. In the navigation scenario, we use 9000 labels for all tasks. In the opposite behaviors scenario, we use 1000 for Window Close, 3000 for Drawer Close, 5000 for Faucet Open, Faucet Close and Window Open, 7000 for Drawer Open, Door Lock and Door Unlock. Also, to reduce the impact of preference-based RL, we additionally add oracle versions of SA-RL and PA-AD, which uses the ground-truth rewards of the targeted task.

We use the same experimental settings (i.e., hyper-parameters, neural networks) concerning reward learning for all methods. We quantitatively evaluate all methods by comparing the success rate of final manipulation, which is well-defined in Meta-world (Yu et al., 2020) for the opposite behaviors scenario, and we rigorously design for the navigation scenario. As in most existing research (Zhang et al., 2020; 2021; Sun et al., 2022), we consider using state attacks with $\ell_\infty$ norm in our experiments, and we report the mean and standard deviation across ten runs for all experiments. We also provide detailed hyper-parameter settings and implementation details in Appendix F.

## 5.2 MANIPULATION ON DRL AGENTS

We study the effectiveness of our method compared to adversarial attack algorithms, which are adapted to our setting with minimal changes. Specifically, we construct two different scenarios on various simulated robotic manipulation tasks. Each victim agent is well-trained for a specific manipulation task.

**Scenarios on Navigation.** In this scenario, we expect the robotic arm to reach a target coordinates instead of completing the original task. Figure 4 shows the training curves of baselines and our method on eight manipulation tasks. It shows that the performance of PALM surpasses that of the baselines by a large margin based on preference labels. To eliminate the influence of preference-based RL and further demonstrate the advantages of PALM, we additionally train the baseline methods with the ground-truth reward function and denote them as "oracle". We notice that the performance of SA-RL (oracle) greatly improves on several tasks over the preference-based version. However, PALM still outperforms SA-RL with oracle rewards on most tasks. These results demonstrate that PALM enables the agent to efficiently learn adversarial policy with human preferences. We also observe that PA-AD is incapable of mastering manipulation, even using the ground-truth rewards.

---

[1] https://github.com/rll-research/BPref
[2] https://github.com/umd-huang-lab/paad_adv_rl

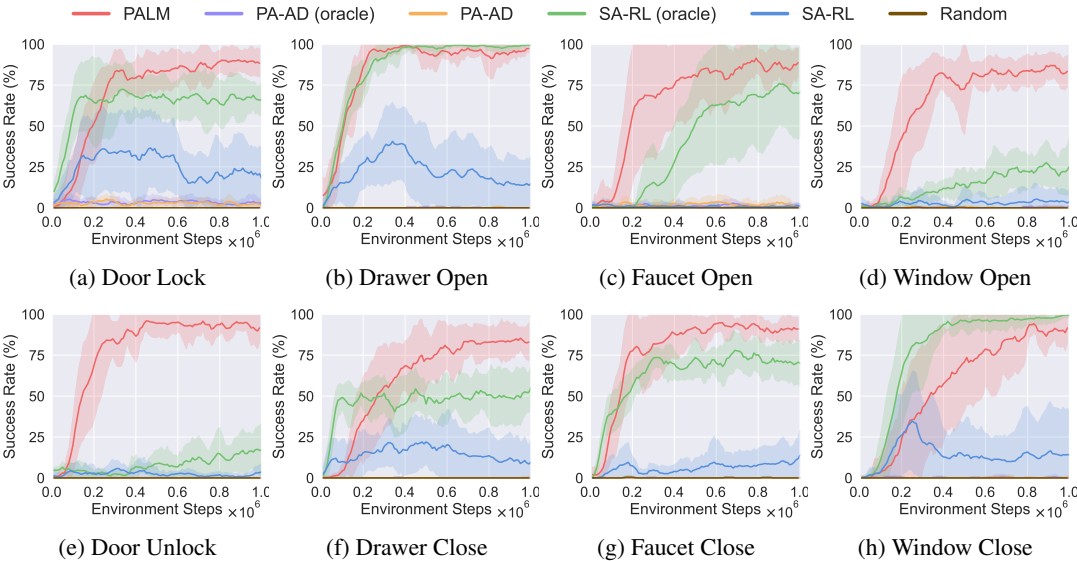

Figure 4: Training curves of different methods on various manipulation tasks in the navigation scenario. The solid line and shaded area denote the mean and the standard deviation of success rate, respectively, over ten runs.

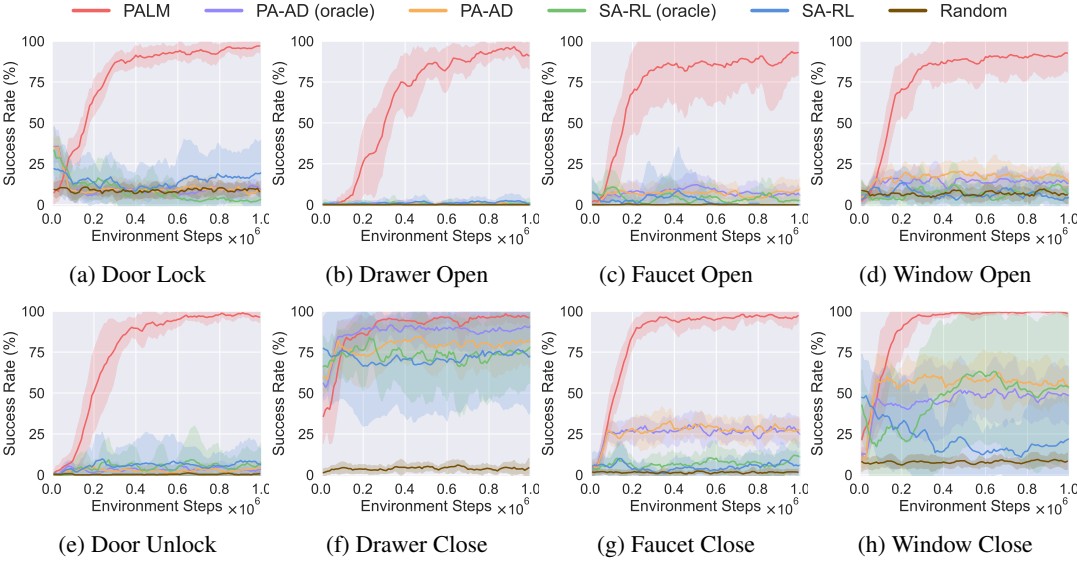

Figure 5: Training curves of all methods on various tasks in the opposite behaviors scenario. The solid line and shaded area denote the mean and the standard deviation of success rate over ten runs.

**Scenarios on Opposite Behaviors.** In the real world, robotic manipulation has good application values. Therefore, we design this scenario to quantitatively evaluate the vulnerability of these agents that masters various manipulation skills. Specifically, we expect each victim to complete the opposite task under the attack of the manipulator. For example, the victim which masters the skill of opening windows will close windows under targeted attack. As shown in Figure 5, PALM presents excellent performance and marginally shows obvious advantages over baseline methods on all tasks. The result again indicates that PALM is effective for a wide range of tasks and can efficiently learn adversarial policy with human preferences.

### 5.3 Manipulation on the Popular Offline RL Agents

In this experiment, we show the vulnerability of offline RL agents and demonstrate PALM can fool them into acting human desired behaviors. As for the implementation, we choose some online

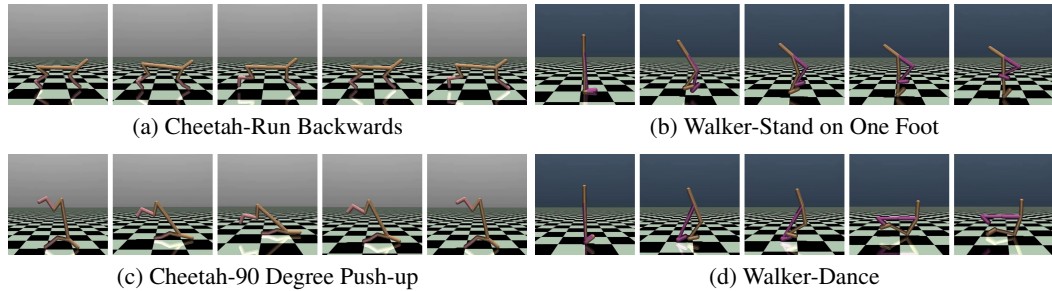

(a) Cheetah-Run Backwards        (b) Walker-Stand on One Foot

(c) Cheetah-90 Degree Push-up        (d) Walker-Dance

Figure 6: Human desired behaviors behaved by the Decision Transformer under the attack of PALM.

models[3] as victims, which are well-trained by official implementation with D4RL. We choose two tasks, Cheetah and Walker, using expert-level Decision Transformer agents as the victims. As shown in Figure 6, Decision Transformer shows exploitable weaknesses and is misled to perform human desired behavior instead of the original task. Specifically, under the adversarial manipulation, the Cheetah agent runs backwards quickly in Figure 6a, and does 90 degree push-up in Figure 6c. The Walker agent stands on one foot for superior balance in Figure 6b, and dances with one leg lifted in Figure 6d. The results show that PALM can manipulate these victims to act behaviors consistent with human preferences and embodied agents are extremely vulnerable to these well-trained adversaries. We hope this experiment can inspire future work on the robustness of offline RL agents and embodied AI.

## 5.4 ABLATION STUDY

**Contribution of Each Component.** We conduct additional experiments to investigate the effect of each component in PALM on Drawer Open, Drawer Close for the navigation scenario and on Faucet Open, Faucet Close for the opposite behavior scenario. PALM contains three critical components: the weight function $h_\omega$, the intention policy $\pi_\theta$, and the combined policy. Table 1 shows that the intention policy plays an essential role in the PALM. As shown in Figure 7, the intention policy can mitigate exploration difficulty caused by the restriction of victim policy and improve the exploration ability of PALM leading to a better adversary. We also observe that the combined policy balances the discrepancy between $\pi_\theta$ and $\pi_{\nu \circ \alpha}$ on the state distribution and improves the adversary's performance. In addition, we can economically train the weighting function to distinguish state importance by formulating the adversary learning as a bi-level optimization. It can further improve the asymptotic performance of PALM. These empirical results show that key ingredients of PALM are fruitfully wed and contribute to the PALM's success.

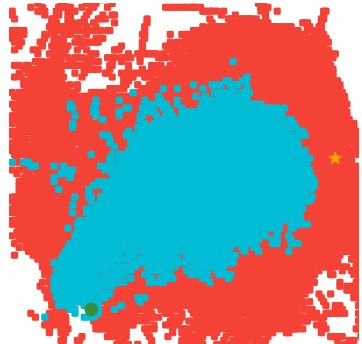

Figure 7: A visualization of the exploration space of PALM (red) and PALM without intention policy (blue). The green point denotes the start and the yellow star denotes the target position.

Table 1: Effects of each component. The success rate on four simulated robotic manipulation tasks from Meta-world. The results are the average success rate across five runs.

| | Drawer Open (Navigation) | Drawer Close (Navigation) | Faucet Open (Opposite) | Faucet Close (Opposite) |
|---|---|---|---|---|
| PALM | 97.2% | 86.8% | 97.2% | 95.5% |
| PALM without $h_\omega$ | 93.2% | 74.8% | 89.4% | 86.2% |
| PALM without $\pi_\theta$ | 13.8% | 11.6% | 0.0% | 4.2% |
| PALM without combination | 38.0% | 22.3% | 51.7% | 71.8% |

---

[3] https://huggingface.co/edbeeching

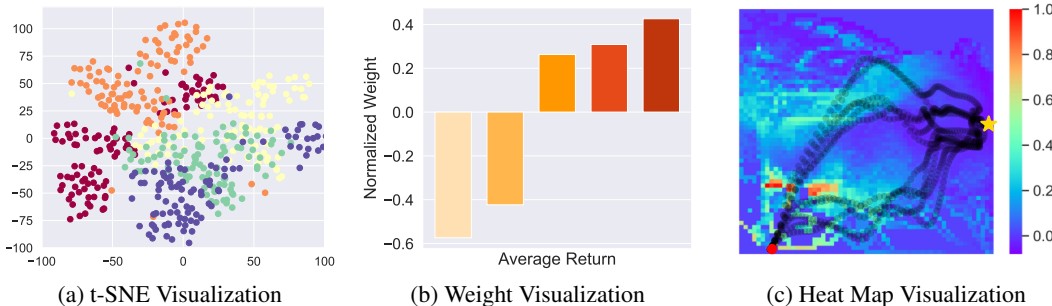

| (a) t-SNE Visualization | (b) Weight Visualization | (c) Heat Map Visualization |

Figure 8: (a) A visualization of the weights of trajectories of different qualities is collected by five different policies. (b) Trajectory weights generated by the weighting function from different policies are extracted and visualized with t-SNE. (c) A heat map showing the weight distribution and the trajectory of the perturbed agent in 2D coordinates. The red point denotes the start position and the yellow star indicates the targeted position.

To verify the restricted exploration problem, we visualize the exploration space of PALM and PALM without intention policy. Figure 7 shows that the intention policy significantly improve the exploration ability of PALM.

**Effects of the Weighting Function.** To further understand the role of the weighting function proposed in Section 4, we conduct experimental data analysis and visualization from multiple perspectives. Five perturbed policies are uniformly sampled with performance increase sequentially before PALM convergence. For each policy, we roll out 100 trajectories and obtain the trajectory weight vectors via the weighting function. By leveraging the technique of t-SNE (van der Maaten & Hinton, 2008), the weight vectors of different policies are visualized in Figure 8a. From the figure, we can clearly observe clear boundaries between the trajectory weights of various policies, suggesting that the weighting function can distinguish trajectories of different qualities. In Figure 8b, the darker color indicates trajectories with higher success rates of manipulation. The result shows that the weighting function gives higher weights to better trajectories for improving the adversarial policy performance. To further illustrate the effect of the weighting function, we present a heat map of the weight distribution in 2D coordinates and annotate part of the trajectories of the perturbed policy. As Figure 8c shows, the weighting function scores the surrounding states in trajectories from the perturbed policy higher, especially in the early stage before reaching the target point.

Extensive experiments are conducted to analyze and discuss the impact of feedback amount and attack budgets on the performance of PALM in the Appendix G.

## 6 CONCLUSION

In this paper, we propose PALM, a preference-based adversarial attack approach against DRL, which can mislead the victim to perform desired behaviors of adversaries. PALM involves an adversary adding imperceptible perturbations on the observations of the victim, an intention policy learned through preference-based RL for better exploration, and a weighting function to identify essential states for the efficient adversarial attack. We analyze the convergence of PALM and prove that PALM converges to critical points under some mild conditions. Empirically, we design two scenarios on several manipulation tasks of Meta-world, and the results demonstrate that PALM outperforms the baselines under the targeted adversarial setting. We further show embodied agents' vulnerability by attacking Decision Transformer on some Mujoco tasks. For future work, we consider: (1) to further improving the attack efficiency by enhancing the utilization efficiency of human preference and (2) extending the observation space of the victim to the high-dimensional inputs, such as images and natural language.

### ETHICS STATEMENT

Preference-based RL provides an effective way to learn agents without a carefully designed reward function. However, learning from human preferences means humans need to provide labeled data which inevitably has biases introducing systematic error. There are possible negative impacts when

malicious people attack other policies using our methods. However, our approach also makes other researchers aware of the vulnerability of policies for AI safety.

REPRODUCIBILITY STATEMENT

The details of experiment settings are provided in Section 4. We provide detailed proofs of theoretical analysis in Appendix D. A more detailed description and implementation setting can be found in Appendix F. Meanwhile, we present the link of our source code and videos in the abstract.

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

## A   THE FULL PROCEDURE OF PALM

**The Combined Policy.** Although $\pi_\theta$ guides adversarial policy learning, the discrepancy between $\pi_\theta$ and $\pi_{\nu \circ \alpha}$ on the state distribution leads to inefficiency. To handle this issue, we design a strategy to construct the behavior policy $\pi$ to collect transitions in our practical implementation. Inspired by Branched rollout (Janner et al., 2019), we combine the intention policy $\pi_\theta$ and the perturbed policy $\pi_{\nu \circ \alpha}$, where $\pi^{1:h} = \pi_{\nu \circ \alpha}^{1:h}$, $\pi^{h+1:H} = \pi_\theta^{h+1:H}$, $h \sim U(0, H)$ and $H$ is task horizon. The combined policy $\pi$ collects data and stores it into the replay buffer during learning.

We provide detailed procedures of our proposed method in Algorithm 1. PALM is implemented based on a popular preference-based RL algorithm PEBBLE (Lee et al., 2021a).

---

**Algorithm 1** PALM

---

**Input:** a fixed victim policy $\pi_\nu$, frequency of human feedback $K$, outer loss updating frequency $M$, task horizon $H$
1: Initialize parameters of $Q_\phi$, $\pi_\theta$, $\widehat{r}_\psi$, $\pi_\alpha$ and $h_\omega$
2: Initialize $\mathcal{B}$ and $\pi_\theta$ with unsupervised exploration
3: Initialize preference data set $\mathcal{D} \leftarrow \emptyset$
4: **for** each iteration **do**
5:      **if** episode is done **then**                                                ▷ Construct the combined policy $\pi$
6:          $h \sim U(0, H)$
7:          $\pi^{1:h} = \pi_{\nu \circ \alpha}^{1:h}$ and $\pi^{h+1:H} = \pi_\theta^{h+1:H}$
8:      **end if**
9:      Take action $a_t \sim \pi$ and collect $s_{t+1}$
10:     Store transition into dataset $\mathcal{B} \leftarrow \mathcal{B} \cup \{(s_t, a_t, \widehat{r}_\psi(s_t), s_{t+1})\}$
11:     **if** iteration % $K == 0$ **then**
12:          **for** each query step **do**                                       ▷ Query preference
13:             Sample pair of trajectories $(\sigma^0, \sigma^1)$
14:             Query preference $y$ from manipulator
15:             Store preference data into dataset $\mathcal{D} \leftarrow \mathcal{D} \cup \{(\sigma^0, \sigma^1, y)\}$
16:          **end for**
17:          **for** each gradient step **do**                                 ▷ Update reward model
18:             Sample batch $\{(\sigma^0, \sigma^1, y)_i\}_{i=1}^n$ from $\mathcal{D}$
19:             Optimize (2) to update $\widehat{r}_\psi$
20:          **end for**
21:      **end if**
22:     **for** each gradient step **do**
23:          Sample random mini-batch transitions from $\mathcal{B}$
24:          Optimize $\pi_\alpha$: minimize (6) with respect to $\alpha$                  ▷ Inner loss optimization
25:      **end for**
26:     **if** iteration % $M == 0$ **then**
27:          Sample random mini-batch transitions from $\mathcal{B}$
28:          Optimize $h_\omega$: minimize (7) with respect to $\omega$                  ▷ Outer loss optimization
29:      **end if**
30:     Update $Q_\phi$ and $\pi_\theta$ according to (3) and (4), respectively.
31: **end for**
**Output:** adversarial policy $\pi_\alpha$

---

## B   DERIVATION OF THE GRADIENT OF THE OUTER-LEVEL LOSS

In this section, we present detailed derivation of the gradient of the outer loss $J_\pi$ with respect to the parameters of the weighting function $\omega$. According to the chain rule, we can derive that

$$
\begin{aligned}
&\nabla_\omega J_\pi(\hat{\alpha}(\omega))\big|_{\omega_t}\\
=&\frac{\partial J_\pi(\hat{\alpha}(\omega))}{\partial \hat{\alpha}(\omega)}\bigg|_{\hat{\alpha}_t} \frac{\partial \hat{\alpha}_t(\omega)}{\partial \omega}\bigg|_{\omega_t}\\
=&\frac{\partial J_\pi(\hat{\alpha}(\omega))}{\partial \hat{\alpha}(\omega)}\bigg|_{\hat{\alpha}_t} \frac{\partial \hat{\alpha}_t(\omega)}{\partial h(\mathbf{s};\omega)}\bigg|_{\omega_t} \frac{\partial h(\mathbf{s};\omega)}{\partial \omega}\bigg|_{\omega_t}\\
=&-\eta_t \frac{\partial J_\pi(\hat{\alpha}(\omega))}{\partial \hat{\alpha}(\omega)}\bigg|_{\hat{\alpha}_t} \sum_{\mathbf{s}\sim\mathcal{B}} \frac{\partial D_{\mathrm{KL}}\left(\pi_{\nu\circ\alpha}(\mathbf{s})\parallel\pi_\theta(\mathbf{s})\right)}{\partial \alpha}\bigg|_{\alpha_t} \frac{\partial h(\mathbf{s};\omega)}{\partial \omega}\bigg|_{\omega_t}\\
=&-\eta_t \sum_{\mathbf{s}\sim\mathcal{B}}\left(\frac{\partial J_\pi(\hat{\alpha}(\omega))}{\partial \hat{\alpha}(\omega)}\bigg|_{\hat{\alpha}_t}^\top \frac{\partial D_{\mathrm{KL}}\left(\pi_{\nu\circ\alpha}(\mathbf{s})\parallel\pi_\theta(\mathbf{s})\right)}{\partial \alpha}\bigg|_{\alpha_t}\right) \frac{\partial h(\mathbf{s};\omega)}{\partial \omega}\bigg|_{\omega_t}.
\end{aligned}
\tag{10}
$$

For brevity of expression, we let:

$$
f(\mathbf{s}) = \frac{\partial J_\pi(\hat{\alpha}(\omega))}{\partial \hat{\alpha}(\omega)}\bigg|_{\hat{\alpha}_t}^\top \frac{\partial D_{\mathrm{KL}}\left(\pi_{\nu\circ\alpha}(\mathbf{s})\parallel\pi_\theta(\mathbf{s})\right)}{\partial \hat{\alpha}}\bigg|_{\alpha_t}.
\tag{11}
$$

The gradient of outer-level optimization loss with respect to parameters $\omega$ is:

$$
\nabla_\omega J_\pi(\hat{\alpha}(\omega))\big|_{\omega_t} = -\eta_t \sum_{\mathbf{s}\sim\mathcal{B}} f(\mathbf{s})\cdot \frac{\partial h(\mathbf{s};\omega)}{\partial \omega}\bigg|_{\omega_t}.
\tag{12}
$$

## C   CONNECTION BETWEEN RSA-MDP AND MDP

**Lemma 1.** *Given a RSA-MDP $\mathcal{M} = (\mathcal{S},\mathcal{A},\mathcal{B},\widehat{\mathcal{R}},\mathcal{P},\gamma)$ and a fixed victim policy $\pi_\nu$, there exists a MDP $\hat{\mathcal{M}} = (\mathcal{S},\hat{\mathcal{A}},\widehat{\mathcal{R}},\widehat{\mathcal{P}},\gamma)$ such that the optimal policy of $\hat{\mathcal{M}}$ is equivalent to the optimal adversary $\pi_\alpha$ in RSA-MDP given a fixed victim, where $\widehat{\mathcal{A}} = \mathcal{S}$ and*

$$
\widehat{\mathcal{P}}(\mathbf{s}'|\mathbf{s},\mathbf{a}) = \sum_{\mathbf{a}\in\mathcal{A}} \pi_\nu(\mathbf{a}|\widehat{\mathbf{a}})\mathcal{P}(\mathbf{s}'|\mathbf{s},\mathbf{a}) \quad \text{for } \mathbf{s},\mathbf{s}'\in\mathcal{S} \text{ and } \widehat{\mathbf{a}}\in\widehat{\mathcal{A}}.
$$

## D   THEORETICAL ANALYSIS AND PROOFS

### D.1   THEOREM 1: CONVERGENCE RATE OF THE OUTER LOSS

**Lemma 2.** *(Lemma 1.2.3 in Nesterov (1998)) If function $f(x)$ is Lipschitz smooth on $\mathbb{R}^n$ with constant L, then $\forall x,y\in\mathbb{R}^n$, we have*

$$
\left|f(y) - f(x) - f'(x)^\top(y-x)\right| \le \frac{L}{2}\|y-x\|^2.
\tag{13}
$$

*Proof.* $\forall x,y\in\mathbb{R}^n$, we have

$$
\begin{aligned}
f(y) &= f(x) + \int_0^1 f'(x+\tau(y-x))^\top(y-x)d\tau\\
&= f(x) + f'(x)^\top(y-x) + \int_0^1 [f'(x+\tau(y-x)) - f'(x)]^\top(y-x)d\tau.
\end{aligned}
\tag{14}
$$

Then we can derive that

$$
\begin{aligned}
\left|f(y) - f(x) - f'(x)^\top(y-x)\right| &= \left|\int_0^1 [f'(x+\tau(y-x)) - f'(x)]^\top(y-x)d\tau\right|\\
&\le \int_0^1 \left|[f'(x+\tau(y-x)) - f'(x)]^\top(y-x)\right|d\tau\\
&\le \int_0^1 \|f'(x+\tau(y-x)) - f'(x)\|\cdot\|y-x\|\,d\tau\\
&\le \int_0^1 \tau L\|y-x\|^2\,d\tau = \frac{L}{2}\|y-x\|^2,
\end{aligned}
\tag{15}
$$

where the first inequality holds for $\left| \int_a^b f(x)dx \right| \leq \int_a^b |f(x)|\, dx$, the second inequality holds for Cauchy-Schwarz inequality, and the last inequality holds for the definition of Lipschitz smoothness. $\qquad\square$

**Theorem 1.** *Suppose $J_\pi$ is Lipschitz-smooth with constant L, the gradient of $J_\pi$ and $\mathcal{L}_{att}$ is bounded by $\rho$. Let the training iterations be T, the inner-level optimization learning rate $\eta_t = \min\{1, \frac{c_1}{T}\}$ for some constant $c_1 > 0$ where $\frac{c_1}{T} < 1$. Let the outer-level optimization learning rate $\beta_t = \min\{\frac{1}{L}, \frac{c_2}{\sqrt{T}}\}$ for some constant $c_2 > 0$ where $c_2 \leq \frac{\sqrt{T}}{L}$, and $\sum_{t=1}^\infty \beta_t \leq \infty, \sum_{t=1}^\infty \beta_t^2 \leq \infty$. The convergence rate of $J_\pi$ achieves*

$$\min_{1 \leq t \leq T} \mathbb{E}\left[ \|\nabla_\omega J_\pi(\alpha_{t+1}(\omega_t))\|^2 \right] \leq \mathcal{O}\left( \frac{1}{\sqrt{T}} \right). \tag{16}$$

*Proof.* First,

$$
\begin{aligned}
&J_\pi(\hat{\alpha}_{t+2}(\omega_{t+1})) - J_\pi(\hat{\alpha}_{t+1}(\omega_t)) \\
&= \{J_\pi(\hat{\alpha}_{t+2}(\omega_{t+1})) - J_\pi(\hat{\alpha}_{t+1}(\omega_{t+1}))\} + \{J_\pi(\hat{\alpha}_{t+1}(\omega_{t+1})) - J_\pi(\hat{\alpha}_{t+1}(\omega_t))\}.
\end{aligned} \tag{17}
$$

Then we separately derive the two terms of (17). For the first term,

$$
\begin{aligned}
&J_\pi(\hat{\alpha}_{t+2}(\omega_{t+1})) - J_\pi(\hat{\alpha}_{t+1}(\omega_{t+1})) \\
&\leq \nabla_{\hat{\alpha}} J_\pi(\hat{\alpha}_{t+1}(\omega_{t+1}))^\top (\hat{\alpha}_{t+2}(\omega_{t+1}) - \hat{\alpha}_{t+1}(\omega_{t+1})) + \frac{L}{2} \|\hat{\alpha}_{t+2}(\omega_{t+1}) - \hat{\alpha}_{t+1}(\omega_{t+1})\|^2 \\
&\leq \|\nabla_{\hat{\alpha}} J_\pi(\hat{\alpha}_{t+1}(\omega_{t+1}))\| \cdot \|\hat{\alpha}_{t+2}(\omega_{t+1}) - \hat{\alpha}_{t+1}(\omega_{t+1})\| + \frac{L}{2} \|\hat{\alpha}_{t+2}(\omega_{t+1}) - \hat{\alpha}_{t+1}(\omega_{t+1})\|^2 \\
&\leq \rho \cdot \|-\eta_{t+1}\nabla_{\hat{\alpha}}\mathcal{L}_{att}(\hat{\alpha}_{t+1})\| + \frac{L}{2} \|-\eta_{t+1}\nabla_{\hat{\alpha}}\mathcal{L}_{att}(\hat{\alpha}_{t+1})\|^2 \\
&\leq \eta_{t+1}\rho^2 + \frac{L}{2}\eta_{t+1}^2\rho^2,
\end{aligned} \tag{18}
$$

where $\hat{\alpha}_{t+2}(\omega_{t+1}) - \hat{\alpha}_{t+1}(\omega_{t+1}) = -\eta_{t+1}\nabla_{\hat{\alpha}}\mathcal{L}_{att}(\hat{\alpha}_{t+1})$, the first inequality holds for Lemma 2, the second inequality holds for Cauchy-Schwarz inequality, the third inequality holds for $\|\nabla_{\hat{\alpha}} J_\pi(\hat{\alpha}_{t+1}(\omega_{t+1}))\| \leq \rho$, and the last inequality holds for $\|\nabla_{\hat{\alpha}}\mathcal{L}_{att}(\hat{\alpha}_{t+1})\| \leq \rho$. It can be proved that the gradient of $\omega$ with respect to $J_\pi$ is Lipschitz continuous and we assume the Lipschitz constant is $L$. Therefore, for the second term,

$$
\begin{aligned}
&J_\pi(\hat{\alpha}_{t+1}(\omega_{t+1})) - J_\pi(\hat{\alpha}_{t+1}(\omega_t)) \\
&\leq \nabla_\omega J_\pi(\hat{\alpha}_{t+1}(\omega_t))^\top (\omega_{t+1} - \omega_t) + \frac{L}{2}\|\omega_{t+1} - \omega_t\|^2 \\
&= -\beta_t \nabla_\omega J_\pi(\hat{\alpha}_{t+1}(\omega_t))^\top \nabla_\omega J_\pi(\hat{\alpha}_{t+1}(\omega_t)) + \frac{L\beta_t^2}{2}\|\nabla_\omega J_\pi(\hat{\alpha}_{t+1}(\omega_t))\|^2 \\
&= -(\beta_t - \frac{L\beta_t^2}{2})\|\nabla_\omega J_\pi(\hat{\alpha}_{t+1}(\omega_t))\|^2,
\end{aligned} \tag{19}
$$

where $\omega_{t+1} - \omega_t = -\beta_t \nabla_\omega J_\pi(\hat{\alpha}_{t+1}(\omega_t))$, and the first inequality holds for Lemma 2. Therefore, (17) becomes

$$J_\pi(\hat{\alpha}_{t+2}(\omega_{t+1})) - J_\pi(\hat{\alpha}_{t+1}(\omega_t)) \leq \eta_{t+1}\rho^2 + \frac{L}{2}\eta_{t+1}^2\rho^2 - (\beta_t - \frac{L\beta_t^2}{2})\|\nabla_\omega J_\pi(\hat{\alpha}_{t+1}(\omega_t))\|^2. \tag{20}$$

Rearranging the terms of (20), we obtain

$$(\beta_t - \frac{L\beta_t^2}{2})\|\nabla_\omega J_\pi(\hat{\alpha}_{t+1}(\omega_t))\|^2 \leq J_\pi(\hat{\alpha}_{t+1}(\omega_t)) - J_\pi(\hat{\alpha}_{t+2}(\omega_{t+1})) + \eta_{t+1}\rho^2 + \frac{L}{2}\eta_{t+1}^2\rho^2. \tag{21}$$

Then, we sum up both sides of (21),

$$
\sum_{t=1}^{T}(\beta_t - \frac{L\beta_t^2}{2}) \|\nabla_\omega J_\pi(\hat{\alpha}_{t+1}(\omega_t))\|^2
$$

$$
\leq J_\pi(\hat{\alpha}_2(\omega_1)) - J_\pi(\hat{\alpha}_{T+2}(\omega_{T+1})) + \sum_{t=1}^{T}(\eta_{t+1}\rho^2 + \frac{L}{2}\eta_{t+1}^2\rho^2) \tag{22}
$$

$$
\leq J_\pi(\hat{\alpha}_2(\omega_1)) + \sum_{t=1}^{T}(\eta_{t+1}\rho^2 + \frac{L}{2}\eta_{t+1}^2\rho^2).
$$

Therefore,

$$
\begin{aligned}
&\min_{1\leq t\leq T} \mathbb{E}\left[\|\nabla_\omega J_\pi(\hat{\alpha}_{t+1}(\omega_t))\|^2\right] \\
&\leq \frac{\sum_{t=1}^{T}(\beta_t - \frac{L\beta_t^2}{2})\|\nabla_\omega J_\pi(\hat{\alpha}_{t+1}(\omega_t))\|^2}{\sum_{t=1}^{T}(\beta_t - \frac{L\beta_t^2}{2})} \\
&\leq \frac{1}{\sum_{t=1}^{T}(2\beta_t - L\beta_t^2)}\left[2J_\pi(\hat{\alpha}_2(\omega_1)) + \sum_{t=1}^{T}(2\eta_{t+1}\rho^2 + L\eta_{t+1}^2\rho^2)\right] \\
&\leq \frac{1}{\sum_{t=1}^{T}\beta_t}\left[2J_\pi(\hat{\alpha}_2(\omega_1)) + \sum_{t=1}^{T}\eta_{t+1}\rho^2(2 + L\eta_{t+1})\right] \\
&\leq \frac{1}{T\beta_t}\left[2J_\pi(\hat{\alpha}_2(\omega_1)) + T\eta_{t+1}\rho^2(2 + L)\right] \\
&= \frac{2J_\pi(\hat{\alpha}_2(\omega_1))}{T\beta_t} + \frac{\eta_{t+1}\rho^2(2 + L)}{\beta_t} \\
&= \frac{2J_\pi(\hat{\alpha}_2(\omega_1))}{T}\max\{L, \frac{\sqrt{T}}{c_2}\} + \min\{1, \frac{c_1}{T}\}\max\{L, \frac{\sqrt{T}}{c_2}\}\rho^2(2 + L) \\
&\leq \frac{2J_\pi(\hat{\alpha}_2(\omega_1))}{c_2\sqrt{T}} + \frac{c_1\rho^2(2 + L)}{c_2\sqrt{T}} \\
&= \mathcal{O}\left(\frac{1}{\sqrt{T}}\right),
\end{aligned} \tag{23}
$$

where the second inequality holds according to (22), the third inequality holds for $\sum_{t=1}^{T}(2\beta_t - L\beta_t^2) \geq \sum_{t=1}^{T}\beta_t$. $\qquad\square$

### D.2 THEOREM 2: CONVERGENCE OF THE INNER LOSS

**Lemma 3.** *(Lemma A.5 in Mairal (2013)) Let $(a_n)_{n\geq 1}, (b_n)_{n\geq 1}$ be two non-negative real sequences such that the series $\sum_{n=1}^{\infty} a_n$ diverges, the series $\sum_{n=1}^{\infty} a_n b_n$ converges, and there exists $C > 0$ such that $|b_{n+1} - b_n| \leq Ca_n$. Then, the sequence $(b_n)_{n\geq 1}$ converges to 0.*

**Theorem 2.** *Suppose $J_\pi$ is Lipschitz-smooth with constant L, the gradient of $J_\pi$ and $\mathcal{L}_{att}$ is bounded by $\rho$. Let the training iterations be T, the inner-level optimization learning rate $\eta_t = \min\{1, \frac{c_1}{T}\}$ for some constant $c_1 > 0$ where $\frac{c_1}{T} < 1$. Let the outer-level optimization learning rate $\beta_t = \min\{\frac{1}{L}, \frac{c_2}{\sqrt{T}}\}$ for some constant $c_2 > 0$ where $c_2 \leq \frac{\sqrt{T}}{L}$, and $\sum_{t=1}^{\infty}\beta_t \leq \infty$, $\sum_{t=1}^{\infty}\beta_t^2 \leq \infty$. $\mathcal{L}_{att}$ achieves*

$$
\lim_{t\to\infty} \mathbb{E}\left[\|\nabla_\alpha \mathcal{L}_{att}(\alpha_t; \omega_t)\|^2\right] = 0. \tag{24}
$$

*Proof.* First,

$$
\begin{aligned}
&\mathcal{L}_{att}(\alpha_{t+1}; \omega_{t+1}) - \mathcal{L}_{att}(\alpha_t; \omega_t) \\
&= \{\mathcal{L}_{att}(\alpha_{t+1}; \omega_{t+1}) - \mathcal{L}_{att}(\alpha_{t+1}; \omega_t)\} + \{\mathcal{L}_{att}(\alpha_{t+1}; \omega_t) - \mathcal{L}_{att}(\alpha_t; \omega_t)\}.
\end{aligned} \tag{25}
$$

For the first term in (25),

$$
\mathcal{L}_{\text{att}}(\alpha_{t+1}; \omega_{t+1}) - \mathcal{L}_{\text{att}}(\alpha_{t+1}; \omega_t)
$$

$$
\leq \nabla_\omega \mathcal{L}_{\text{att}}(\alpha_{t+1}; \omega_t)^\top (\omega_{t+1} - \omega_t) + \frac{L}{2} \|\omega_{t+1} - \omega_t\|^2 \tag{26}
$$

$$
= - \beta_t \nabla_\omega \mathcal{L}_{\text{att}}(\alpha_{t+1}; \omega_t)^\top \nabla_\omega J_\pi(\alpha_{t+1}(\omega_t)) + \frac{L\beta_t^2}{2} \|\nabla_\omega J_\pi(\alpha_{t+1}(\omega_t))\|^2 .
$$

where $\omega_{t+1} - \omega_t = -\beta_t \nabla_\omega J_\pi(\alpha_{t+1}(\omega_t))$, and the first inequality holds according to Lemma 2. For the second term in (25),

$$
\mathcal{L}_{\text{att}}(\alpha_{t+1}; \omega_t) - \mathcal{L}_{\text{att}}(\alpha_t; \omega_t)
$$

$$
\leq \nabla_\alpha \mathcal{L}_{\text{att}}(\alpha_t; \omega_t)^\top (\alpha_{t+1} - \alpha_t) + \frac{L}{2} \|\alpha_{t+1} - \alpha_t\|^2
$$

$$
= - \eta_t \nabla_\alpha \mathcal{L}_{\text{att}}(\alpha_t; \omega_t)^\top \nabla_\alpha \mathcal{L}_{\text{att}}(\alpha_t; \omega_t) + \frac{L\eta_t^2}{2} \|\nabla_\alpha \mathcal{L}_{\text{att}}(\alpha_t; \omega_t)\|^2 \tag{27}
$$

$$
= - (\eta_t - \frac{L\eta_t^2}{2}) \|\nabla_\alpha \mathcal{L}_{\text{att}}(\alpha_t; \omega_t)\|^2 .
$$

where $\alpha_{t+1} - \alpha_t = -\eta_t \nabla_\alpha \mathcal{L}_{\text{att}}(\alpha_t; \omega_t)$, and the first inequality holds according to Lemma (2). Therefore, (25) becomes

$$
\mathcal{L}_{\text{att}}(\alpha_{t+1}; \omega_{t+1}) - \mathcal{L}_{\text{att}}(\alpha_t; \omega_t)
$$

$$
\leq - \beta_t \nabla_\omega \mathcal{L}_{\text{att}}(\alpha_{t+1}; \omega_t)^\top \nabla_\omega J_\pi(\alpha_{t+1}(\omega_t)) + \frac{L\beta_t^2}{2} \|\nabla_\omega J_\pi(\alpha_{t+1}(\omega_t))\|^2 \tag{28}
$$

$$
- (\eta_t - \frac{L\eta_t^2}{2}) \|\nabla_\alpha \mathcal{L}_{\text{att}}(\alpha_t; \omega_t)\|^2 .
$$

Taking expectation of both sides of (28) and rearranging the terms, we obtain

$$
\eta_t \mathbb{E}\left[\|\nabla_\alpha \mathcal{L}_{\text{att}}(\alpha_t; \omega_t)\|^2\right] + \beta_t \mathbb{E}\left[\|\nabla_\omega \mathcal{L}_{\text{att}}(\alpha_{t+1}; \omega_t)\| \cdot \|\nabla_\omega J_\pi(\alpha_{t+1}(\omega_t))\|\right]
$$

$$
\leq \mathbb{E}\left[\mathcal{L}_{\text{att}}(\alpha_t; \omega_t)\right] - \mathbb{E}\left[\mathcal{L}_{\text{att}}(\alpha_{t+1}; \omega_{t+1})\right] + \frac{L\beta_t^2}{2} \mathbb{E}\left[\|\nabla_\omega J_\pi(\alpha_{t+1}(\omega_t))\|^2\right] \tag{29}
$$

$$
+ \frac{L\eta_t^2}{2} \mathbb{E}\left[\|\nabla_\alpha \mathcal{L}_{\text{att}}(\alpha_t; \omega_t)\|^2\right] .
$$

Summing up both sides of (29) from $t = 1$ to $\infty$,

$$
\sum_{t=1}^\infty \eta_t \mathbb{E}\left[\|\nabla_\alpha \mathcal{L}_{\text{att}}(\alpha_t; \omega_t)\|^2\right] + \sum_{t=1}^\infty \beta_t \mathbb{E}\left[\|\nabla_\omega \mathcal{L}_{\text{att}}(\alpha_{t+1}; \omega_t)\| \cdot \|\nabla_\omega J_\pi(\alpha_{t+1}(\omega_t))\|\right]
$$

$$
\leq \mathbb{E}\left[\mathcal{L}_{\text{att}}(\alpha_1; \omega_1)\right] - \lim_{t \to \infty} \mathbb{E}\left[\mathcal{L}_{\text{att}}(\alpha_{t+1}; \omega_{t+1})\right] + \sum_{t=1}^\infty \frac{L\beta_t^2}{2} \mathbb{E}\left[\|\nabla_\omega J_\pi(\alpha_{t+1}(\omega_t))\|^2\right] \tag{30}
$$

$$
+ \sum_{t=1}^\infty \frac{L\eta_t^2}{2} \mathbb{E}\left[\|\nabla_\alpha \mathcal{L}_{\text{att}}(\alpha_t; \omega_t)\|^2\right]
$$

$$
\leq \sum_{t=1}^\infty \frac{L(\eta_t^2 + \beta_t^2)\rho^2}{2} + \mathbb{E}\left[\mathcal{L}_{\text{att}}(\alpha_1; \omega_1)\right] \leq \infty,
$$

where the second inequality holds for $\sum_{t=1}^\infty \eta_t^2 \leq \infty$, $\sum_{t=1}^\infty \beta_t^2 \leq \infty$, $\|\nabla_\alpha \mathcal{L}_{\text{att}}(\alpha_t; \omega_t)\| \leq \rho$, $\|\nabla_\omega J_\pi(\alpha_{t+1}(\omega_t))\| \leq \rho$. Since

$$
\sum_{t=1}^\infty \beta_t \mathbb{E}\left[\|\nabla_\omega \mathcal{L}_{\text{att}}(\alpha_{t+1}; \omega_t)\| \cdot \|\nabla_\omega J_\pi(\alpha_{t+1}(\omega_t))\|\right] \leq L\rho \sum_{t=1}^\infty \beta_t \leq \infty. \tag{31}
$$

Therefore, we have

$$
\sum_{t=1}^\infty \eta_t \mathbb{E}\left[\|\nabla_\alpha \mathcal{L}_{\text{att}}(\alpha_t; \omega_t)\|^2\right] < \infty. \tag{32}
$$

Since $|(\|a\| + \|b\|)(\|a\| - \|b\|)| \leq \|a + b\|\|a - b\|$, we can derive that

$$
\begin{aligned}
&\left| \mathbb{E}\left[ \|\nabla_\alpha \mathcal{L}_{\text{att}}(\alpha_{t+1}; \omega_{t+1})\|^2 \right] - \mathbb{E}\left[ \|\nabla_\alpha \mathcal{L}_{\text{att}}(\alpha_t; \omega_t)\|^2 \right] \right| \\
=&\left| \mathbb{E}\left[ \left( \|\nabla_\alpha \mathcal{L}_{\text{att}}(\alpha_{t+1}; \omega_{t+1})\| + \|\nabla_\alpha \mathcal{L}_{\text{att}}(\alpha_t; \omega_t)\| \right) + \left( \|\nabla_\alpha \mathcal{L}_{\text{att}}(\alpha_{t+1}; \omega_{t+1})\| - \|\nabla_\alpha \mathcal{L}_{\text{att}}(\alpha_t; \omega_t)\| \right) \right] \right| \\
\leq&\mathbb{E}\left[ \left| \|\nabla_\alpha \mathcal{L}_{\text{att}}(\alpha_{t+1}; \omega_{t+1})\| + \|\nabla_\alpha \mathcal{L}_{\text{att}}(\alpha_t; \omega_t)\| \right| \left| \|\nabla_\alpha \mathcal{L}_{\text{att}}(\alpha_{t+1}; \omega_{t+1})\| - \|\nabla_\alpha \mathcal{L}_{\text{att}}(\alpha_t; \omega_t)\| \right| \right] \\
\leq&\mathbb{E}\left[ \|\nabla_\alpha \mathcal{L}_{\text{att}}(\alpha_{t+1}; \omega_{t+1}) + \nabla_\alpha \mathcal{L}_{\text{att}}(\alpha_t; \omega_t)\| \cdot \|\nabla_\alpha \mathcal{L}_{\text{att}}(\alpha_{t+1}; \omega_{t+1}) - \nabla_\alpha \mathcal{L}_{\text{att}}(\alpha_t; \omega_t)\| \right] \\
\leq&\mathbb{E}\left[ \left( \|\nabla_\alpha \mathcal{L}_{\text{att}}(\alpha_{t+1}; \omega_{t+1})\| + \|\nabla_\alpha \mathcal{L}_{\text{att}}(\alpha_t; \omega_t)\| \right) \|\nabla_\alpha \mathcal{L}_{\text{att}}(\alpha_{t+1}; \omega_{t+1}) - \nabla_\alpha \mathcal{L}_{\text{att}}(\alpha_t; \omega_t)\| \right] \\
\leq&2L\rho \mathbb{E}\left[ \|(\alpha_{t+1}, \omega_{t+1}) - (\alpha_t, \omega_t)\| \right] \\
\leq&2L\rho \eta_t \beta_t \mathbb{E}\left[ \|(\nabla_\alpha \mathcal{L}_{\text{att}}(\alpha_t; \omega_t), \nabla_\omega J_\pi(\alpha_{t+1}(\omega_t)))\| \right] \\
\leq&2L\rho \eta_t \beta_t \sqrt{\mathbb{E}\left[ \|\nabla_\alpha \mathcal{L}_{\text{att}}(\alpha_t; \omega_t)\|^2 \right] + \mathbb{E}\left[ \|\nabla_\omega J_\pi(\alpha_{t+1}(\omega_t))\|^2 \right]} \\
\leq&2L\rho \eta_t \beta_t \sqrt{2\rho^2} \\
\leq&2\sqrt{2}L\rho^2 \eta_t \beta_t.
\end{aligned}
\tag{33}
$$

Since $\sum_{t=1}^\infty \eta_t = \infty$, according to Lemma 3, we have

$$
\lim_{t \to \infty} \mathbb{E}\left[ \|\nabla_\alpha \mathcal{L}_{\text{att}}(\alpha_t; \omega_t)\|^2 \right] = 0.
\tag{34}
$$

$\square$

# E  DETAILS OF PBRL

In this section, we present details of the scripted teacher and preference collection. It is a crucial part of the PbRL, and PALM follows these settings as Lee et al. (2021a).

**Scripted Teacher.** To evaluate the performance systemically, a useful way is to consider a scripted teacher that provides preferences between a pair of agent's trajectory segments according to the oracle reward function. Leveraging the preference labels from the human teacher is ideal, while it is hard to evaluate algorithms quantitatively and quickly. Specifically, the scripted teacher can immediately provide ground truth rewards based on the state $\mathbf{s}$ and action $\mathbf{a}$. It is a function designed to approximate the human's intention.

**Preference Collection.** During training, we need to query human preference labels at regular intervals. It samples a batch of segment pairs and calculates the cumulative reward of each segment with rewards provided by the scripted teacher. For a specific segment pair, human prefers the segment with a larger cumulative reward. The segment with a larger cumulative reward is labelled with 1, and the smaller one is labelled with 0. As for the computational cost, we assume that $M$ preference labels are required, the segment length is $N$ in a run, and the time complexity is $\mathcal{O}(MN)$. However, it is negligible compared with adversary training, which involves complex gradient computation.

# F  EXPERIMENTAL DETAILS

In this section, we provide a concrete description of our experiments and detailed hyper-parameters of PALM. For each run of experiments, we run on GeForce RTX 2080 Ti GPUs for training.

## F.1  TASKS

We conduct experiments on eight robotic manipulation tasks from Meta-world (Yu et al., 2020) in phase one and two locomotion tasks from Mujoco (Todorov et al., 2012) in phase two. The tasks we used are:

**Meta-world**

- Door Lock: An agent controls a simulated Sawyer arm to lock the door.

- Door Unlock: An agent controls a simulated Sawyer arm to unlock the door.

- Drawer Open: An agent controls a simulated Sawyer arm to open the drawer to a target position.

- Drawer Close: An agent controls a simulated Sawyer arm to close the drawer to a target position.

- Faucet Open: An agent controls a simulated Sawyer arm to open the faucet to a target position.

- Faucet Close: An agent controls a simulated Sawyer arm to close the faucet to a target position.

- Window Open: An agent controls a simulated Sawyer arm to open the window to a target position.

- Window Close: An agent controls a simulated Sawyer arm to close the window to a target position.

**Mujoco**

- Half Cheetah: A 2-dimensional robot with nine links and eight joints aims to learn to run forward (right) as fast as possible.

- Walker: A 2-dimensional two-legged robot aims to move in the forward (right).

### F.2 HYPER-PARAMETERS SETTING

We choose PEBBLE as For SA-RL (Zhang et al., 2021), we keep the same parameter setting with parameter and use the same neural network structure as ours. The detailed hyper-parameters of SA-RL is shown in Table 3. For PA-AD (Sun et al., 2022), all hyper-parameters are the same as those of SA-RL.

Table 2: Hyper-parameters of PALM for adversary training.

| Hyper-parameter | Value | Hyper-parameter | Value |
|---|---|---|---|
| Number of layers | 3 | Hidden units of each layer | 256 |
| Learning rate | 0.0003 | Batch size | 1024 |
| Length of segment | 50 | Number of reward functions | 3 |
| Frequency of feedback | 5000 | Feedback batch size | 128 |
| Adversarial budget | 0.1 | $(\beta_1, \beta_2)$ | $(0.9, 0.999)$ |

Table 3: Hyper-parameters of SA-RL for adversary training.

| Hyper-parameter | Value | Hyper-parameter | Value |
|---|---|---|---|
| Number of layers | 3 | Hidden units of each layer | 256 |
| Learning rate | 0.00005 | Mini-Batch size | 32 |
| Length of segment | 50 | Number of reward functions | 3 |
| Frequency of feedback | 5000 | Feedback batch size | 128 |
| Adversarial budget | 0.1 | Entropy coefficient | 0.0 |
| Clipping parameter | 0.2 | Discount $\gamma$ | 0.99 |
| GAE lambda | 0.95 | KL divergence target | 0.01 |

### F.3 VICTIM SETTING

Our experiment consists of two phases. In the first phase, we use various simulated robotic manipulation tasks from Meta-world. We have two OpenAI Gym MuJoCo continuous control environments in the second phase.

**Meta-world.** We train the victims on Meta-world by utilizing SAC (Haarnoja et al., 2018) with the original fully connected neural network as policy. Detailed hyper-parameters are shown in Table 4.

Table 4: Hyper-parameters of SAC for victim training.

| Hyper-parameter | Value | Hyper-parameter | Value |
|---|---|---|---|
| Number of layers | 3 | Initial temperature | 0.1 |
| Hidden units of each layer | 256 | Optimizer | Adam |
| Learning rate | 0.0001 | Critic target update freq | 2 |
| Discount $\gamma$ | 0.99 | Critic EMA $\tau$ | 0.005 |
| Batch size | 1024 | $(\beta_1, \beta_2)$ | $(0.9, 0.999)$ |
| Steps of unsupervised pre-training | 9000 | Discount $\gamma$ | 0.99 |

**Mujoco.** We directly utilize the well-trained model for demonstrating the vulnerability of the Decision Transformer. Specifically, we use the Cheetah agent[4] and the Walker agent[5] with expert-level.

## G    EXTENSIVE EXPERIMENTS

**Impact of Feedback Amount.** We investigate the performance of PALM with different preference labels. Table 5 shows the results of all methods with various numbers of labels: $\{3000, 5000, 7000, 9000\}$ on Drawer Open for the navigation scenario and $\{1000, 3000, 5000, 7000\}$ on Faucet Close for the opposite behavior scenario. From the experimental results in Table 5, we conclude that providing sufficient human feedback learns a strong adversary and stabilizes the attack success rate. We notice that the performance of PALM improves with the increase of human preference labels, indicating that the number of preference labels has an essential impact on adversary learning. However, the performance of SA-RL and PA-AD is poor even with enough human feedback and PA-AD completely fails in navigation scenario. The reason is that the exploration space of these two methods is limited by the fixed victim policy, while PALM achieves better exploration by introducing an intention policy.

Table 5: Success rate with various amount of preference labels on Drawer Open for the navigation scenario and Faucet Close for the opposite behavior scenario. We report the mean and the standard deviation of the success rate over 30 episodes.

| Environment | Feedback | PALM (ours) | PA-AD | SA-RL |
|---|---|---|---|---|
| **Drawer Open** (navigation) | 3000 | $65.7\% \pm 37.1\%$ | $0.0\% \pm 0.0\%$ | $8.3\% \pm 13.2\%$ |
| | 5000 | $86.7\% \pm 18.1\%$ | $0.0\% \pm 0.0\%$ | $21.3\% \pm 18.9\%$ |
| | 7000 | $95.7\% \pm 13.6\%$ | $0.0\% \pm 0.0\%$ | $28.0\% \pm 28.1\%$ |
| | 9000 | $97.0\% \pm 6.9\%$ | $0.0\% \pm 0.0\%$ | $13.0\% \pm 18.5\%$ |
| **Faucet Close** (opposite behavior) | 1000 | $69.7\% \pm 35.2\%$ | $16.7\% \pm 9.4\%$ | $2.0\% \pm 6.0\%$ |
| | 3000 | $79.0\% \pm 16.2\%$ | $29.0\% \pm 14.0\%$ | $6.0\% \pm 11.7\%$ |
| | 5000 | $95.3\% \pm 9.2\%$ | $21.3\% \pm 12.8\%$ | $3.3\% \pm 12.7\%$ |
| | 7000 | $95.3\% \pm 7.6\%$ | $22.7\% \pm 12.4\%$ | $4.0\% \pm 7.1\%$ |

**Impact of Different Attack Budgets.** We also analyze the effect of the attack budget. For further understanding, we conduct additional experiments with various attack budgets: $\{0.05, 0.075, 0.1, 0.15\}$ on the Drawer Open and $\{0.02, 0.05, 0.075, 0.1\}$ on Faucet Close for these two scenarios. In the Figure 9, we report the performance of baseline and PALM with various attack budgets. As the experimental results show, the performance of all methods improve as the adversarial budget increases.

---

[4] https://huggingface.co/edbeeching/decision-transformer-gym-halfcheetah-expert
[5] https://huggingface.co/edbeeching/decision-transformer-gym-walker2d-expert

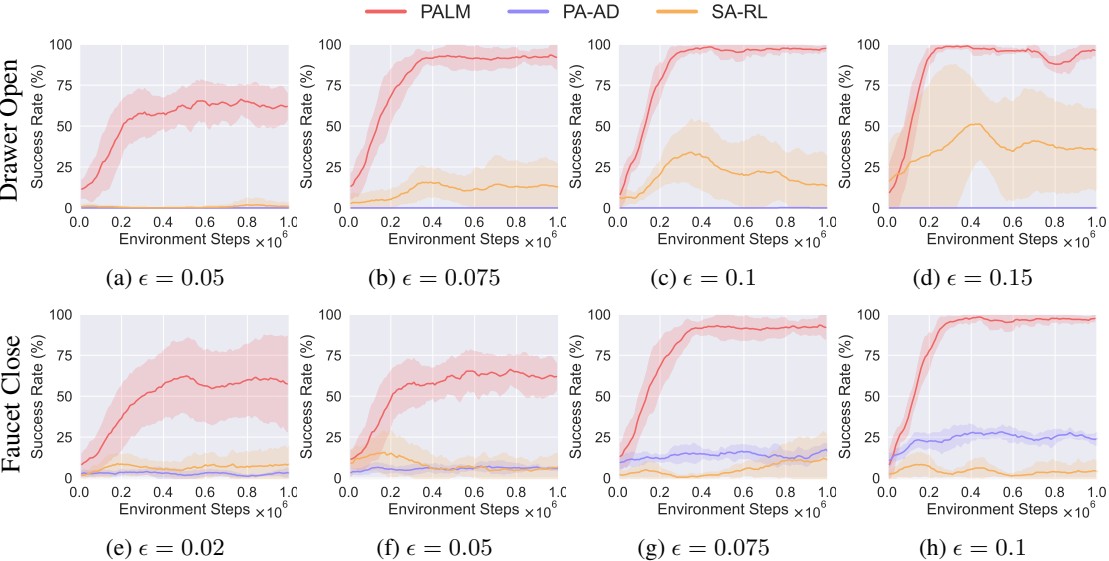

Figure 9: Training curves of success rate with different adversarial budgets on Drawer Open for the navigation scenario and Faucet Close for the opposite behavior scenario. The solid line and shaded area denote the mean and the standard deviation of the success rate across five runs.

