# OpenReview forum: "PALM: Preference-based Adversarial Manipulation against Deep Reinforcement Learning"
_ICLR.cc/2023/Conference — Submitted to ICLR 2023_

### Official Review · Reviewer_x6Lt · 2022-10-23

**Confidence:** 3
**Correctness:** 4
**Technical Novelty And Significance:** 3
**Empirical Novelty And Significance:** 3
**Recommendation:** 6

**Clarity, Quality, Novelty And Reproducibility:**

To allow better assessment of the originality of the paper, it would be useful if the authors could elaborate on how the fact that their approach "emphasizes manipulating victim policy" makes it unique? How far is it from existing approaches? How hard would it be to empower these with the ability of performing targeted attacks (instead of making the agent fail or choose the worst action)?

In terms of clarity, (besides the first weakness listed above), some sentences are not clear. For examples:
- In Section 2, "Unlike previous research, past works focus on making victim policy work fail, while our approach...". Should "Unlike previous research" be removed?
- In Section 3, "The target of GSA-MDP is to solve the optimal adversary pi_alpha*, which enables the victim to achieve the maximum expected return over all states". Why would we want to enable the victim to succeed?
- In Section 3, the context for reward optimization from preferences is not clear at this point. Where will this be used? More context should be given in the section or alternatively, this part could be moved to the next section, where it is used.
- In Section 4, "which indicates the optimal adversarial manipulation without the restriction of the victim".
- In Section 5, "and we rigorously design for the navigation scenario". What do you design?
Also, some of the notation is not explained in the text. For example, in Section 3, \alpha in \pi_{\alpha}, \hat{A} and \hat{P}, the norm p (which norm is used)?

Smaller comments:
- Figure 1 is not referred to in the text.
- Section 3, 3rd line: is reward function -> is the reward function
- Section 4, \tau_t is the transition at time t: if \tau_t ~ B, isn't it a state? what does transition mean?
- Section 4.1. missing parenthesis around 5 (above equation (7))

**Strength And Weaknesses:**

The main strengths of this work are:
1. A novel approach that allows to manipulate deep RL agents towards a certain target (as opposed to simply make the agent fail in its task).
2. Supporting theoretical analysis on the convergence of PALM.

The main weaknesses of this work are:
1. An (intuitive) explanation for the role of the intention policy (beyond the given equations) is lacking. What is the motivation behind this? How does it interplay with the adversarial policy and guide its training?
2. In the experiments, the authors compare PALM to existing algorithms that were not designed to manipulate the victim agent policy, in manipulation tasks. It would be interesting to see what happens if the situation is the other way around. How does PALM fair in experiments settings for which SA-RL and PA-AD are designed?
3. Data is collected following a combined policy that depends on the perturbed policy. It is not clear why/how this can be done in practice. What happens if we cannot control according to which policy data is collected?

**Summary Of The Paper:**

This work proposes the Preference-based Advisersarial Manipulation (PALM) algorithm which performs targeted attacks on deep reinforcement learning agents. This method used human preferences feedback and relies on an intention policy and a weighting function to guide the adversarial policy during training. Empirical results show that PALM outperforms several flavors of the adversarial algorithms SA-RL and PA-AD (which, as opposed to PALM, were not designed to manipulate the victim agent policy).

**Summary Of The Review:**

In summary, the authors present an interesting approach for target attacks on deep reinforcement learning agents. The main weaknesses of this work are that the level of novelty and originality is not fully clear and it also seems that the empirical results showing that the proposed approach outperforms existing algorithms were performed in "unfair" settings.

---

> ### Author Response · Authors · 2022-11-13
> **Response to Reviewer x6Lt (Part 2/2)**
>
> (Continue)
>
> **Q6: "In Section 3, "The target of GSA-MDP is to solve the optimal adversary $\pi_\alpha^\*$, which enables the victim to achieve the maximum expected return over all states". Why would we want to enable the victim to succeed?"**
>
> > **A6**: It is worth to emphasize that the reward function $\hat{R}$ is consistent with human preferences. When the victim policy performs the human desired behaviors under attack from the adversary, it can obtain a high return. Therefore, the victim can achieve the maximum expected return over all states under the attacks of the optimal adversarial policy.
>
> **Q7: "In Section 3, the context for reward optimization from preferences is not clear at this point. Where will this be used? More context should be given in the section or alternatively, this part could be moved to the next section, where it is used."**
>
> > **A7**: Reward learning from human preferences is used in Section 4 for intention policy learning. Thanks for your suggestion and we rearrange this part in the revised draft.
>
> **Q8: "In Section 4, "which indicates the optimal adversarial manipulation without the restriction of the victim"."**
>
> > **A8**: Here it means the intention policy provides unrestricted exploration and guidance for targeted attack during the adversarial policy learning. We will revise this sentence in the paper.
>
> **Q9: "In Section 5, "and we rigorously design for the navigation scenario". What do you design?"**
>
> > **A9**: We mainly design a selection method of the target point, we choose it randomly from one of the half-spaces which does not contain the origin goal.
>
> **Q10: "Also, some of the notation is not explained in the text. For example, in Section 3, $\alpha$ in $\pi_{\alpha}$, $\hat{A}$ and $\hat{P}$, the norm p (which norm is used)?"**
>
> > **A10**: $\alpha$ are the parameters of $\pi_{\alpha}$. $\hat{A}$, $\hat{P}$ are the action space and transition probability of the adversary, respectively. We will make these clear in the revised paper.
>
> **Q11: "Smaller comments: 1\) Figure 1 is not referred to in the text.
> 2\) Section 3, 3rd line: is reward function -> is the reward function
> 3\) Section 4, \tau_t is the transition at time t: if $\tau_t \sim B$, isn't it a state? what does transition mean?
> 4\) Section 4.1. missing parenthesis around 5 (above equation (7))"**
>
> > **A11**: Thanks for your advice. Transition $\tau_t$ is a tuple $(s_t, a_t, r_t, s_{t+1})$ contains current state $s_t$, agent's action $a_t$, reward $r_t$ and next state $s_{t+1}$. We will revise these in the updated version.

---

> ### Author Response · Authors · 2022-11-13
> **Response to Reviewer x6Lt (Part 1/2)**
>
> We thank reviewer x6Lt for your positive support. We will provide point-wise explainations to your questions.
>
> **Q1: "An (intuitive) explanation for the role of the intention policy (beyond the given equations) is lacking. What is the motivation behind this? How does it interplay with the adversarial policy and guide its training?"**
>
> > **A1**: 1\) If we directly learn an adversary to perturb the observation of the victim policy within a small budget, the exploration will be limited by the victim policy. Therefore, it will be quite difficult to manipulate the victim policy to perform human desired behaviors far from its original behaviors. We introduce an intention policy which has large exploration space to tackle this issue. We will add a visualization of the exploration space in Figure 7 in the revised version. Figure 7 also demonstrates that the intention policy significantly improves the exploration ablitity of PALM.
> > 2\) "guide" means the intention policy is the learning target of the adversarial policy. The adversarial policy is trained by minimizing the re-weighted KL divergence between it and the intention policy. We will supplement the motivation of introducing an intention policy in the draft.
>
> **Q2: "In the experiments, ... It would be interesting to see what happens if the situation is the other way around. How does PALM fair in experiments settings for which SA-RL and PA-AD are designed?"**
>
> > **A2**: Changing PALM for the untargeted settings of SA-RL and PA-AD is insignificant because PALM will degenerate to SA-RL with PbRL in that case.
>
> **Q3: "Data is collected following a combined policy that depends on the perturbed policy. It is not clear why/how this can be done in practice. What happens if we cannot control according to which policy data is collected?"**
>
> > **A3**: 1\) In our implementation, we first sample $h \sim U(0, H)$ where $H$ is the task horizon. We use the perturbed policy $\pi_{\nu \circ \alpha}$ to collect data from time step $1$ to $h$ and finally receive $s_{h+1}$. Then we use the intention policy to collect data starting from $s_{h+1}$ from time step $h+1$ to $H$.
> > 2\) If we cannot control which policy is used, there may have two problems. If the perturbed policy collect almost all data, PALM will be restricted to the small exploration space of the victim policy. If the intention policy collect almost all data, PALM will suffer from the out-of-distribution (OOD) problem. Specifically, PALM uses Equation (6) (old version) which contains the Q-function to evaluate the perturbed policy. If the Q-function is trained by the data from the intention policy through Equation (4) (old version), using Equation (6) (old version) will suffer from OOD.
>
> **Q4: "To allow better assessment of the originality of the paper, it would be useful if the authors could elaborate on how the fact that their approach "emphasizes manipulating victim policy" makes it unique? How far is it from existing approaches? How hard would it be to empower these with the ability of performing targeted attacks (instead of making the agent fail or choose the worst action)?"**
>
> > **A4**: Thank you for your question.
> > 1\) Most previous researches focus on the untargeted adversarial attack against DRL agents. Under untargeted attacks, agents have various ways to fail, i.e., for a victim agent which walks in a straight line, turning left and right are both failures. However, it is challenging to induce the victim to perform specific behaviors far from its original goal.
> > 2\) Compared with existing methods, PALM differs in that we leverage an intention policy to provide unrestricted exploration and guidance for targeted attack. And the intention policy is learned via preference-based RL to avoid reward engineering.
> > 3\) To achieve targeted attacks, the first problem is restricted exploration. The victim policy is fixed and has a limited exploration space. To clearly demonstrate the effectiveness of the intention policy, we will add a visualization of the exploration space of PALM and PALM without intention policy in our revised paper (to be shown in Figure 7). So we introduce the intention policy to solve this problem. The second problem is that the reward function of the targeted task is difficult to design. We avoid reward engineering by utilizing preference-based RL, which learns a well-performing policy with human preferences.
>
> **Q5: "In Section 2, "Unlike previous research, past works focus on making victim policy work fail, while our approach...". Should "Unlike previous research" be removed?"**
>
> > **A5**: Thank you for your suggestion and we will update it in the revised version.

---

> ### Author Response · Authors · 2022-11-17
> **A gentle reminder**
>
> Dear Reviewer x6Lt,
>
> We would like to know whether we have addressed your concerns. If you have any further questions, we are happy to discuss.

---

> > ### Comment · Reviewer_x6Lt · 2022-11-23
> > **Response to the authors**
> >
> > Dear Authors,
> >
> > Thank you for your detailed responses and explantations, as well as the updates performed in the manuscript.
> > One remaining comment is that I still think that the limitation highlighted in Q3 and addressed in A3 should be outlined clearly in the paper.

---

> > > ### Author Response · Authors · 2022-11-23
> > > **Response to Reviewer x6Lt**
> > >
> > > Dear Reviewer x6Lt,
> > >
> > > Thank you for your advice. We will outline these clearly in the next version.

---

### Official Review · Reviewer_bmdX · 2022-10-25

**Confidence:** 4
**Correctness:** 2
**Technical Novelty And Significance:** 2
**Empirical Novelty And Significance:** 2
**Recommendation:** 5

**Clarity, Quality, Novelty And Reproducibility:**

As far as I can tell the idea of using preference based RL for adversarial attacks is novel.

**Strength And Weaknesses:**

Strengths:
* The novelty of the idea of using Preference based RL for adversarial attacks.
it is often hard to define what a successful attack is formally, but you know it when you see it, so this could be a promising direction

* The intention policy seems promising as an approach for solving exploration problems in adversarial attack domains

Weaknesses:
* The paper seems to not test the core claim: that preference Based RL is useful for adversarial attacks

Most of the tasks are tasks for which a hard-coded reward function does exist(maybe all of the tasks, the setup is a bit unclear).  As such, they can't be effectively used to show that preference based RL is useful for adversarial attacks since the problem of specifying the task or solved.

The counterargument to the above presented by the authors (that it would be hard to specify directly specify reward functions for tasks like Go), is contradicted by  Alpha Zero which worked off of a hard-coded human reward, and learned an even more well shaped reward in terms of the value function.

* It seems that all of the empirical improvements come from the intention policy rather than the preference based RL

-- The preference based model is betting other models which have the ground-truth reward function.  This shouldn't happen if the only change is to use preference based RL, since in the best case preference based RL will reduce to the oracle model.  Thus the only other change (the intention policy), is the probable cause for the difference, but this is not framed as the core contribution of the paper.  If it were, it should be analyzed outside the preference based RL setting, since it does not depend on that.

**Summary Of The Paper:**

This paper studies the problem of using preference based RL to adversarially attack Deep RL policies.  They propose a novel intention policy as a way to stabilize training which leads to improved performance in this regime

**Summary Of The Review:**

I'm recommending weak rejection of this paper since it seems like the empirical results do not justify the conclusion, due to the intention policy confounding experiments on preference based RL, and preference based RL not being properly evaluated on its own.

---

> ### Author Response · Authors · 2022-11-13
> **Response to Reviewer bmdX**
>
> We thank reviewer bmdX for careful and valuable comments and we will respond to your concerns point by point.
>
> **Q1: "The paper seems to not test the core claim: that preference based RL is useful for adversarial attacks ... As such, they can't be effectively used to show that preference based RL is useful for adversarial attacks since the problem of specifying the task or solved."**
>
> > **A1**: 1\) We use tasks with ground-truth reward functions to conveniently evaluate the effectiveness of our method. Following previous preference-based RL works [1,2,3,4], we use a scripted teacher to label preferences. During training, the intention policy cannot access the ground-truth reward but only preference labels.
> > 2\) Alpha Zero utilizes MCTS which can attain high complexity when the state space is large. Although it works well, it takes plenty of computational resources to train an agent which is quite expensive and is not a general solution for most people. However, preference-based RL is an easier way to learn a well-shaped reward function with human preferences. On the other hand, Alpha Zero can only handle discrete states, but preference-based RL is suitable in both continuous and discrete tasks.
>
> **Q2: "It seems that all of the empirical improvements come from the intention policy rather than the preference based RL. The preference based model is betting other models which have the ground-truth reward function. This shouldn't happen if the only change is to use preference based RL, since in the best case preference based RL will reduce to the oracle model. Thus the only other change (the intention policy), is the probable cause for the difference, but this is not framed as the core contribution of the paper. If it were, it should be analyzed outside the preference based RL setting, since it does not depend on that."**
>
> > **A2**: 1\) To achieve targeted attacks, the first problem is exploraron difficulty caused by restriction of victim policy. So we introduce the intention policy to solve this issue. 2\) The second problem is that the reward function of the targeted task is challenging to design. We avoid reward engineering by utilizing preference-based RL, which learns a well-performing policy with human preferences. Therefore, intention policy and preference-based RL are crucial for PALM.
>
> [1] Christiano et al. Deep reinforcement learning from human preferences. NeurIPS 2017.
>
> [2] Lee et al. Pebble: Feedback-efficient interactive reinforcement learning via relabeling experience and unsupervised pre-training. ICML 2021.
>
> [3] Park et al. SURF: Semi-supervised reward learning with data augmentation for feedback-efficient preference-based reinforcement learning. ICLR 2022.
>
> [4] Liang et al. Reward uncertainty for exploration in preference-based reinforcement learning. ICLR 2022.

---

> > ### Comment · Reviewer_bmdX · 2022-11-25
> > **Response To Authors**
> >
> > Thank you for your detailed response.
> >
> > >We use tasks with ground-truth reward functions to conveniently evaluate the effectiveness of our method. Following previous preference-based RL works [1,2,3,4], we use a scripted teacher to label preferences. During training, the intention policy cannot access the ground-truth reward but only preference labels.
> >
> > I agree this is the proper way to evaluate a preference-based learning algorithm is functioning properly (which is what all of the papers cited in the authors response are showing).  However, this is not the proper way to tell if a preference based learning algorithm is useful for a specific task (which is what the paper under review is trying to show).
> >
> > >Alpha Zero utilizes MCTS which can attain high complexity when the state space is large. Although it works well, it takes plenty of computational resources to train an agent which is quite expensive and is not a general solution for most people. However, preference-based RL is an easier way to learn a well-shaped reward function with human preferences. On the other hand, Alpha Zero can only handle discrete states, but preference-based RL is suitable in both continuous and discrete tasks.
> >
> > This is all beside the point, and different than what was argued in the paper.  What was originally argued is that learning a well-shaped reward function is the bottleneck for these problems, using go specifically as an example, when it is not the bottleneck in practice.  Most RL techniques learn a value function somewhere, PPO, Alpha Go, Q Learning, all have something like it, and that is effectively "learning the shaping".  So if that is a key argument you would need to be comparing to one of those traditional methods to show that the preference-based comparisons are actually having an effect on simplifying the exploration problem.  However, the results are confounded as to make that hard to determine.
> >
> > A side-point, preference learning doesn't usually guarantee that the learned reward function is well shaped, so it's unclear why the learned reward function would have any exploration benefit over the original.
> >
> > >1) To achieve targeted attacks, the first problem is exploration difficulty caused by restriction of victim policy. So we introduce the intention policy to solve this issue. 2) The second problem is that the reward function of the targeted task is challenging to design. We avoid reward engineering by utilizing preference-based RL, which learns a well-performing policy with human preferences. Therefore, intention policy and preference-based RL are crucial for PALM.
> >
> > I understand that those are the two problems that the authors asserted exist.  This seems like a restatement of the authors original position without acknowledging the core critique of Q2.
> >
> > To restate the core critique: the benefits of the system come from #1 and not #2.  Though everything is framed in terms of #2, it seems unnecessary.  However, all of the experiments are confounded so neither me nor the authors can argue conclusively for one side or the other without the necessary ablation.  Since this abblation has not been done I think the central claims of the paper are unjustified, and will leave my score unchanged.

---

> ### Author Response · Authors · 2022-11-17
> **A gentle reminder**
>
> Dear Reviewer bmdX,
>
> We would like to know whether we have addressed your concerns. If you have any further questions, we are happy to discuss.

---

### Official Review · Reviewer_kjbK · 2022-10-27

**Confidence:** 4
**Correctness:** 3
**Technical Novelty And Significance:** 2
**Empirical Novelty And Significance:** 3
**Recommendation:** 5

**Clarity, Quality, Novelty And Reproducibility:**

Clarity: can be improved a lot.

Quality: experimentation is well-executed but the quality of the presentation and of the evidence in favor of the proposed method is sub-par.

Novelty: first preference-based method for targeted adversarial attacks on Deep RL agents, though the influence of the preference-based component is not properly quantified.

Reproducibility: the code and some videos are available through anonymized links.


**Strength And Weaknesses:**

### Strengths

Figures and the detailed algorithm are well-designed and help understand some aspects of the work.

Using an intention policy to explore the support of states that lead to maximizing the adversarial human reward is an interesting idea. It would be nice to show how much suboptimal exploration would hinder the same approach without the intention policy.

Experimentation seems to be solid and the proposed method performs better than other benchmarked methods in most tasks.

Having code and videos available is a very nice plus.

### Weaknesses

The title should mention human-in-the-loop in my opinion.

The related work is not complete at all. It is missing its core: targeted attacks for Deep RL.
A reference comes to mind: Targeted Attacks on Deep Reinforcement Learning Agents through Adversarial Modifications (Hussenot et al, 2020).
While this can be fixed, I find it puzzling that the authors have not found it useful or took the time to include at least a few references in that direction, and better a complete review of the literature. I am willing to give authors the benefit of doubt but this needs to be addressed before I am able to recommend for acceptance.

On the problem setup: 1) “generalized state-adversarial” is not a great name, should be “rewarded state-adversarial” instead, or something that better illustrates that there is a reward component involved, 2) “victim policy” is an offensive terminology, use ‘perturbed’/‘target’ instead maybe?

It is unclear whether the reward learning method proposed is a contribution or not.

$\pi_{\nu\circ\alpha}$ is not defined (though one can figure out what it means).

On the method: the connection to SAC and soft-Q-learning should be mentioned, it is not clear whether it is a contribution or not in the current state, and this choice has no motivation (e.g. continuous control).

On the objective:
Is it necessary to reweigh the KL-divergence? This is a modification of the standard that should be better motivated. E.g. via an ablation study.

I do not get why $\omega$ is learned via this implicit parameterization. Could it simply be fixed and seen as a hyperparameter? In other words, this is an additional source of complexity (bi-level optimization) of the algorithm and its inclusion should be better motivated.
Another point is that the nature of \pi_\alpha should be clarified sooner in the paper. How does it operate on the observations?? Is $\alpha$ kept fixed?

I think there should be evidence (i.e. empirical, via an ablation study) that the intention policy is needed.
Also, is it not possible to have an epsilon-greedy switch between perturbed and intention instead of the switching heuristic proposed?

Drawbacks include that the intention policy has to be learned semi-online (part of the episode is played according to the intention policy), not simply through off-policy learning. Experiments showing it is not possible would be welcome.

The main drawback for me is that the writing is not clear enough on key parts of the work. Some super important details (weighting, bi-level optimization, and crucially *the nature of the adversarial modifications*) are hard to grasp or even missing from the current state of the paper.

There are no details on preference collection! This is absolutely crucial to the work.

Nit: the videos are not very impressive, the adversarial command always consists of performing the inverse of the sequence of actions from the policy. Adding other settings would better convey the usefulness of the approach.


**Summary Of The Paper:**

The paper presents a novel method for targeted adversarial attacks on Deep Reinforcement Learning algorithms by modifying agent observations. It learns an adversarial reward function via the preferences of a human-in-the-loop, and uses it to guide an intention policy that itself generates behaviors the adversarial policy eventually learns to reproduce by perturbing agent observations.


**Summary Of The Review:**

The work features an interesting idea and convincing experimental results.

Though, I feel that the work is not ready for publication yet, and that a proper literature review, improvements to the writing, clarifications and additional ablations would make the work stand out much more (see my detailed comments).
I cannot recommend this work for acceptance in its current state.

---

> ### Author Response · Authors · 2022-11-13
> **Response to Reviewer kjbK (Part 2/2)**
>
> (Continue)
>
> **Q11: "I think there should be evidence (i.e. empirical, via an ablation study) that the intention policy is needed."**
>
> > **A11:** Ablation study results in the following table show that intention policy plays an essential role in PALM. Intuitively, the intention policy is needed because the exploration space of the victim policy is limited. By introducing it, PALM has an unlimited exploration space so that it can manipulate the agent to desired goals. We will add a visualization of the exploration space in Figure 7 in the revised version. Figure 7 also demonstrates that the intention policy significantly improves the exploration ablitity of PALM.
> >
> > |                               | Drawer Open | Drawer Close | Faucet Open | Faucet Close |
> > | :---------------------------: | :---------: | :----------: | :---------: | :----------: |
> > | PALM                          | 94.7%       | 99.0%        | 95.4%       | 96.8%        |
> > | PALM without intention policy | 0.0%        | 69.8%        | 0.0%        | 3.0%         |
>
> **Q12: "Also, is it not possible to have an epsilon-greedy switch between perturbed and intention instead of the switching heuristic proposed?"**
>
> > **A12**: 1) Epsilon-greedy strategy is a trade-off between exploration and exploitation. This is not the case of our method because exploitation does not exist in PALM.
> > 2) Branched rollout is a common technique used in model-based RL [5], which uses a learned model to roll out from a branch point. We use the random switch to obtain a similar rollout method.
> > 3) Due to the limited exploration space of the victim policy, PALM achieves better exploration by switching to the intention policy.
>
> **Q13: "Drawbacks include that the intention policy has to be learned semi-online, not simply through off-policy learning."**
>
> > **A13**: We conduct experiments to study the effect of the combined policy. The results in the following table show that semi-online learning is beneficial for PALM. If the intention policy is learned in the off-policy paradigm, PALM will suffer from the out-of-distribution (OOD) problem. Specifically, PALM uses Equation (6) (old version) which contains the Q-function to evaluate the perturbed policy. If the Q-function is trained by the data from the intention policy through Equation (4) (old version), using Equation (6) (old version) will suffer from OOD.
> >
> > |                          | Drawer Open | Drawer Close | Faucet Open | Faucet Close |
> > | :----------------------: | :---------: | :----------: | :---------: | :----------: |
> > | PALM                     | 94.7%       | 99.0%        | 95.4%       | 96.8%        |
> > | PALM without combination | 87.0%       | 87.2%        | 30.9%       | 76.2%        |
>
> **Q14: "The main drawback for me is that the writing is not clear enough on key parts of the work. Some super important details (weighting, bi-level optimization, and crucially the nature of the adversarial modifications) are hard to grasp or even missing from the current state of the paper."**
>
> > **A14**: We will reorganize the method section and make it clearer in the revised vesion.
>
> **Q15: "There are no details on preference collection! This is absolutely crucial to the work."**
>
> > **A15**: Thanks for your advice, and we will add details on collecting preferences in Section 5.1. Specifically, we use a scripted teacher to annotate preferences on segment pairs as previous preference-based RL methods [4,6,7,8]. The scripted teacher is assumed to have full expertise on the task and can precisely reflect ground-truth reward function by providing preferences.
>
> **Reference**
>
> [1] Gleave et al. Adversarial policies: Attacking deep reinforcement learning. ICLR 2020.
>
> [2] Zhang et al. Robust reinforcement learning on state observations with learned optimal adversary. ICLR 2021.
>
> [3] Sun et al. Who is the strongest enemy? Towards optimal and efficient evasion attacks in deep RL. ICLR 2022.
>
> [4] Lee et al. Pebble: Feedback-efficient interactive reinforcement learning via relabeling experience and unsupervised pre-training. ICML 2021.
>
> [5] Janner et al. When to trust your model: Model-based policy optimization. NeurIPS 2019.
>
> [6] Christiano et al. Deep reinforcement learning from human preferences. NeurIPS 2017.
>
> [7] Park et al. SURF: Semi-supervised reward learning with data augmentation for feedback-efficient preference-based reinforcement learning. ICLR 2022.
>
> [8] Liang et al. Reward uncertainty for exploration in preference-based reinforcement learning. ICLR 2022.

---

> > ### Comment · Reviewer_kjbK · 2022-11-15
> > **Response to authors' response**
> >
> > A1: On the one hand, authors mention that “preference-based” means “human-in-the-loop”, which I disagree with: preferences can either be defined computationally or by a human.
> > On the other hand, I do not see any experiment with a human in the loop, since authors mention in A15 that they use scripted teachers. I am confused about this, and I think either 1) actual human-in-the-loop experiments are conducted or 2) the mention of “human-in-the-loop” is avoided in the paper.
> >
> > A2: Thanks. I am pleased to see a proper related work on target attacks in RL in the updated version. I still have an issue with the claims though: it is said that “past works focus on making the victim policy fail, while our approach emphasizes manipulating victim policy” which is factually untrue (e.g. Hussenot et al). Please fix.
> >
> > A3: Thanks.
> >
> > A4: I understand that the “victim” terminology is standard but would comprehension be hindered by replacing it with “target”? In my opinion, it would not. Note that this debate is not influencing my appreciation of the paper.
> >
> > A5: The fact that the reward learning method presented in the text is not a contribution is still not clear enough. This should be stated without ambiguity in section 4.1, including a citation.
> >
> > A6: It is still not explained how states are perturbed, other than the limit imposed on state modification. Are authors masking the observation? Perturbing with Gaussian noise? This is an important detail that is missing.
> >
> > A7: Good.
> >
> > A8: Thanks for running these ablation studies. Please include these in the updated paper.
> >
> > A9: I get that if parameterized you need this bi-level optimization. My point was about whether this parameterization is needed or not. This is somehow answered by the ablation studies of A8, but this point should be made clear when introducing the parameterization for state importance. Authors could also choose to discuss a version of their algorithm that uses uniform state importance, which would be less performant but with lower training computational cost overall.
> >
> > A10: Still unclear, see response to A6.
> >
> > A11: Very nice ablation study. This is something to include in the paper as well, to motivate the intention policy. It should be made clear that this is crucial.
> >
> > A12: The authors misunderstood my point, but it was not correct. I was thinking about switching between intention and victim with a given, fixed probability.
> >
> > A13: Thanks for the ablation study.
> >
> > A14: I think there is still lots of room for improvement on that aspect.
> >
> > A15: There is still no detail as to the nature of the scripted teacher (indeed, it is mentioned that the setup is similar to an existing paper, but my point is that these details should be added at least to the Appendix). Nor any form of analysis of the complexity and/or computational cost overhead introduced by the preference collection.
> > This is key to make sense of the comparison to other methods, which I am not sure is fair in the current state of the paper.
> >
> > *General comment*:
> >
> > Authors did a good job on the ablation studies proposed and on the level of details added to the paper.
> >
> > There are still several points that bother me:
> > * there is no mention of the computational cost of the preference-based component, which is crucial for a fair comparison with baselines
> > * some important details are still missing (e.g. the nature of state perturbations)
> > * some claims are not precise enough (human-in-the-loop without human involving experiments)
> > * the quality of the writing is not excellent, notably in the parts added during the response phase (this can be addressed after the paper is accepted, but this has to be mentioned)
> >
> > On the other hand, I will upgrade my score to better reflect the state of the paper.

---

> > > ### Author Response · Authors · 2022-11-17
> > > **Response to Reviewer kjbK (Part 2/2)**
> > >
> > > (Continue)
> > >
> > > **Q12: "The authors misunderstood my point, but it was not correct. I was thinking about switching between intention and victim with a given, fixed probability."**
> > >
> > > > **A12**: The intent policy should be executed continuously for better exploration. In this way, it can keep the consistency of the behaviour policy. On the other hand, it avoids introducing a new hyperparameter (i.e., switch probability).
> > >
> > > **Q14: "I think there is still lots of room for improvement on that aspect."**
> > >
> > > > **A14**: Thank you for your advice. We would like to improve our writing and make our methods clearer.
> > >
> > > **Q15: "There is still no detail as to the nature of the scripted teacher (indeed, it is mentioned that the setup is similar to an existing paper, but my point is that these details should be added at least to the Appendix). Nor any form of analysis of the complexity and/or computational cost overhead introduced by the preference collection. This is key to make sense of the comparison to other methods, which I am not sure is fair in the current state of the paper."**
> > >
> > > > **A15**: A scripted teacher, can provide preferences between trajectory segments according to the ground truth reward. It calculates the cumulative reward of segments and compares the values. The segment with the larger cumulative reward is labelled with 1, and the smaller one is labelled with 0. We assume there are $M$ preference labels required, and the segment length is $N$ in a run. Then, the time complexity is $\mathcal{O}(MN)$, which is negligible compared with the adversary training. We will add these details to the Appendix of the new version.
> > >
> > > **Reference**
> > >
> > > [1] Christiano et al. Deep reinforcement learning from human preferences. NeurIPS 2017.
> > >
> > > [2] Lee et al. Pebble: Feedback-efficient interactive reinforcement learning via relabeling experience and unsupervised pre-training. ICML 2021.
> > >
> > > [3] Park et al. SURF: Semi-supervised reward learning with data augmentation for feedback-efficient preference-based reinforcement learning. ICLR 2022.
> > >
> > > [4] Liang et al. Reward uncertainty for exploration in preference-based reinforcement learning. ICLR 2022.

---

> > > > ### Comment · Reviewer_kjbK · 2022-11-29
> > > > **Response to authors**
> > > >
> > > > I thank the authors for their comments. I am still convinced that the paper is not ready for presentation at ICLR due to unclear or unsupported statements.

---

> > > ### Author Response · Authors · 2022-11-17
> > > **Response to Reviewer kjbK (Part 1/2)**
> > >
> > > We thank Reviewer kjbK for the response and provide our point-wise response below.
> > >
> > > **Q1: "On the one hand, authors mention that “preference-based” means “human-in-the-loop”, which I disagree with: preferences can either be defined computationally or by a human. On the other hand, I do not see any experiment with a human in the loop, since authors mention in A15 that they use scripted teachers. I am confused about this, and I think either 1) actual human-in-the-loop experiments are conducted or 2) the mention of “human-in-the-loop” is avoided in the paper."**
> > >
> > > > **A1**: For systemically performance evaluation, using the preferences of the scripted teacher is a common setting in PbRL [1,2,3,4]. And there is a hidden assumption that scripted teachers can approximate humans [4].
> > >
> > > **Q2: Thanks. I am pleased to see a proper related work on target attacks in RL in the updated version. I still have an issue with the claims though: it is said that “past works focus on making the victim policy fail, while our approach emphasizes manipulating victim policy” which is factually untrue (e.g. Hussenot et al). Please fix.**
> > >
> > > > **A2**: Thank you for your suggestion. We will revise this sentence in the latest version.
> > >
> > > **Q4: "I understand that the “victim” terminology is standard but would comprehension be hindered by replacing it with “target”? In my opinion, it would not. Note that this debate is not influencing my appreciation of the paper."**
> > >
> > > > **A4**: We think it is better to use the standard terminology to avoid potential confusion.
> > >
> > > **Q5: "The fact that the reward learning method presented in the text is not a contribution is still not clear enough. This should be stated without ambiguity in section 4.1, including a citation."**
> > >
> > > > **A5**: Thank you for your suggestion. We will revise it in the latest version.
> > >
> > > **Q6: "It is still not explained how states are perturbed, other than the limit imposed on state modification. Are authors masking the observation? Perturbing with Gaussian noise? This is an important detail that is missing."**
> > >
> > > > **A6**: We don't calculate $\tilde{s}$ directly during manipulating. Instead, the adversary $\pi_\alpha$ produces a Gaussian noise $\Delta$ with $\ell_\infty(\Delta)$ less than 1. We project $s+\Delta$ to $\mathcal{B}(s)$ through $\tilde{s} = s + \Delta*\epsilon$. In this way, the adversary perturbs the state $s$ into $\tilde{s}$, and the victim takes action according to $\tilde{s}$. We will add these details in the new version.
> > >
> > > **A8: "Thanks for running these ablation studies. Please include these in the updated paper."**
> > >
> > > > **A8**: We will add these results in the latest version.
> > >
> > > **Q9: "I get that if parameterized you need this bi-level optimization. My point was about whether this parameterization is needed or not. This is somehow answered by the ablation studies of A8, but this point should be made clear when introducing the parameterization for state importance. Authors could also choose to discuss a version of their algorithm that uses uniform state importance, which would be less performant but with lower training computational cost overall."**
> > >
> > > > **A9**: The results from the ablation study of $h_\omega$ show that the weighting function $h_\omega$ can improve the performance of $\pi_\alpha$. Parameterization is an effective method to optimize $h_\omega$, which makes good use of adversary's dependency on $h_\omega$. In addition, training for $h_\omega$ is not expensive. We optimize the $\omega$ every 3000 iterations (in Algorithm 1 Line 26-28), which is negligible compared with the adversary training. Optimizing $h_\omega$ does not increase the computational cost too much, but it can improve the performance of PALM.
> > >
> > > **Q11: "Very nice ablation study. This is something to include in the paper as well, to motivate the intention policy. It should be made clear that this is crucial."**
> > >
> > > > **A11**: Thank you for your advice. We will make it clearer in the latest version.
> > >
> > > (Citations included in Part 2)

---

> > > ### Author Response · Authors · 2022-11-18
> > > **A Reminder to Reviewer kjbK**
> > >
> > > Dear Reviewer kjbK,
> > >
> > > We have updated the manuscript and supplementary.

---

> ### Author Response · Authors · 2022-11-13
> **Response to Reviewer kjbK (Part 1/2)**
>
> We thank reviewer kjbK for careful and constructive comments and we will respond to your concerns point by point.
>
> **Q1: "The title should mention human-in-the-loop in my opinion."**
>
> > **A1**: Thank you for your suggestion. Preference in our title refers to human preferences, which also means human-in-the-loop.
>
> **Q2: "The related work is not complete at all. It is missing its core: targeted attacks for Deep RL...better a complete review of the literature."**
>
> > **A2**: Thanks for your advice. We will add a detailed review of targeted attacks against DRL agents in the revised version.
>
> **Q3: ""generalized state-adversarial" is not a great name, should be “rewarded state-adversarial” instead, or something that better illustrates that there is a reward component involved."**
>
> > **A3**: Thank you for your advice and we will update "generalized state-adversarial" to "rewarded state-adversarial" in the revised version.
>
> **Q4: "“victim policy” is an offensive terminology, use ‘perturbed’/‘target’ instead maybe?"**
>
> > **A4**: This is a conventional usage in the context of adversarial RL [1,2,3]. Perturbed policy means the victim policy perturbed by the adversary [3].
>
> **Q5: "It is unclear whether the reward learning method proposed is a contribution or not."**
>
> > **A5**: As for the reward learning method, we directly utilize the preference-based RL algorithm PEBBLE [4]. It is worth to emphasize that one of our contributions is to apply the technique of PbRL to avoid reward engineering in the targeted attack.
>
> **Q6: "$\pi_{\nu \circ \alpha}$ is not defined (though one can figure out what it means)."**
>
> > **A6**: $\pi_{\nu \circ \alpha}$ is the perturbed policy which takes action following $\pi_\nu$ according to the perturbed state $\tilde{s}$ from $\pi_{\alpha}$. We will add the definition of $\pi_{\nu \circ \alpha}$ in the revised version.
>
> **Q7: "On the method: the connection to SAC and soft-Q-learning should be mentioned, it is not clear whether it is a contribution or not in the current state, and this choice has no motivation (e.g. continuous control)."**
>
> > **A7**: Thanks for the advice. SAC is a direct application in our method and we can apply any RL algorithms. We use SAC because it is a robust continuous control method and achieves great performance on many tasks. We will add the connection to SAC and soft Q-learning as well as the motivation in our paper.
>
> **Q8: "On the objective: Is it necessary to reweigh the KL-divergence? This is a modification of the standard that should be better motivated. E.g. via an ablation study."**
>
> > **A8**: We conduct an additional ablation study to examine the effectiveness of the weighting function. The results in the following table demonstrate that the weighting function can improve the performance of PALM.
> >
> > |                         | Drawer Open | Drawer Close | Faucet Open | Faucet Close |
> > | :---------------------: | :---------: | :----------: | :---------: | :----------: |
> > | PALM                    | 94.7%       | 99.0%        | 95.4%       | 96.8%        |
> > | PALM without $h_\omega$ | 85.7%       | 93.7%        | 85.9%       | 82.7%        |
>
> **Q9: "I do not get why $\omega$ is learned via this implicit parameterization. Could it simply be fixed and seen as a hyperparameter?"**
>
> > **A9**: It is infeasible to directly optimize the parameters of the weighting function due to the lack of a labelled dataset. We notice that the performance of $\pi_\alpha$ depends on $h_\omega$ from Equation (5) (old version). Therefore, we consider evaluating the weighting function according to performance of $\pi_\alpha$ and optimize $\omega$ using Equation (6) (old version).
>
> **Q10: "Another point is that the nature of $\pi_\alpha$ should be clarified sooner in the paper. How does it operate on the observations? Is $\alpha$ kept fixed?"**
>
> > **A10**: PALM aims to learn an adversarial policy $\pi_\alpha$ whose parameters $\alpha$ are optimized through Equation (5) (old version). Specifically, the adversarial policy $\pi_\alpha$ perturbs the state $s$ into $\tilde{s} \in$ {$\tilde{s} | \parallel s-\tilde{s} \parallel_\infty \le \epsilon$}, where $\epsilon$ is the attack budget. More details can be found in the Section 3 and we will make this clearer.
>
> (Citations included in Part 2)

---

### Official Review · Reviewer_2Rf7 · 2022-11-02

**Confidence:** 3
**Correctness:** 4
**Technical Novelty And Significance:** 4
**Empirical Novelty And Significance:** 4
**Recommendation:** 6

**Clarity, Quality, Novelty And Reproducibility:**

* **Clarity**: for the general paper outline see weaknesses above. language has some flaws (e.g. "we wonder that" followed by a question in the intro is unnatural) but this is not a major issue.

* **Quality**: the work itself seems very strong.

* **Novelty/Originality**: While not 100% familiar with the state of the art, I do not think targeted attacks on RL agents have been attempted before. The problem is novel, and the solution is complex and original.

* **Reproducibility** is satisfying. Putting the code on anonymous github is very helpful. I did not try to reproduce the results but it would be an easy check.

**Strength And Weaknesses:**

**Strengths**: The addressed problem is challenging. The proposed solution is detailed and convincing: it consists of multiple steps, leverages several optimization methods, and is expressed in a complete formalism. This solution also seems largely novel and does not simply extend past work. The experimental setup seems complete and the results very strong. Online demos help grasping better while the paper accomplishes on practical examples.

**Weaknesses**: One weak point is clarity. As a reader very familiar with adversarial attacks but less so with Reinforcement learning, understanding the main points of the paper was difficult. Combining standard RL formalism, intention policy, adversarial policy, victim, multiple rewards and optimizations, etc. lead to a heavy formalism. Given the complexity of the work technicality is necessary, and the authors do provide a diagram of the pipeline, but I think clarity could be improved. Parts of the method remain unclear to me: for instance, I am not sure whether the intention policy is learned before, or jointly to the weighing function and adversarial policy (Figure 2 says jointly, but the method seem to imply before).

Another point I am not sure about is the attack budget: it is mentioned in appendix, and the introduction says that perturbations are imperceptible, but the actual method does not make clear how that budget is handled or why it is necessary. Overall I'd suggest having a full part dedicated to the outline of the method, keeping technical aspects to a minimum, and moving the rest (theoretical results, learning rates, etc) somewhere else. More detailed diagrams for subparts of the pipeline would also be useful.

Another issue is the limited amount of motivation the authors provide for their work. One sentence in the introduction says that studying adversarial attacks is "crucial", but why some threat models rather than others? With an attack like PALM what could an attacker concretely do in the real world? Motivating or defining informally specific aspects of the work could be useful as well. What is an "imperceptible perturbation"? What imperceptible means on image or speech is clear, but on RL states it is harder to grasp. When it comes to theoretical results, they come with so-called "mild" conditions. Are those conditions typically met? Having convincing answer to those questions would make the paper stronger.

A final (more minor) point: in absence of competing targeted attacks the authors make straightforward modifications to untargeted attacks to get baselines. That this would lead to weak baselines was predictable, and the comparison isn't too fair. Providing it is better than not, but I do not think it is an essential part of the paper, and it could be moved to an appendix without loss.

**Summary Of The Paper:**

This paper proposes a way to run targeted adversarial attacks against Reinforcement Learning agents. The threat model consist in modifying the state perceived by the agent in order to trigger specific actions. The method is to both find an optimal "intention policy" that achieves the target behavior, then try to match it with an adversarial policy in which states are perturbed. The latter is formulated as bi-level optimization. The authors provide a training procedure for that problem and show that under some conditions it will converge to a critical point. Experimental results show that this attack can achieve the target behavior against multiple RL agents, and outperforms multiple baselines by large margins. Online examples demonstrate the attack's success in Meta-world.

**Summary Of The Review:**

The paper provides what seems to be a strong work with very good results. I think a subpar paper outline focusing to much on technicality and not enough on motivations undermine the work. I tend to prefer the work accepted, but it would benefit from some rewriting.

---

> ### Author Response · Authors · 2022-11-13
> **Response to Reviewer 2Rf7 (Part 2/2)**
>
> (Continue)
>
> **Q7: "When it comes to theoretical results, they come with so-called "mild" conditions. Are those conditions typically met?"**
>
> > **A7**: The "mild" conditions include: 1) Lipschitz-smoothness of $J_\pi$, 2) the gradient bound $\rho$ of $J_\pi$ and $\mathcal{L}_{att}$, and 3) learning rate conditions. The first one is a common condition used in theoretical analysis, e.g. Proofs in Adam [9]. We do not need to condition on all data, but some samples are enough, which is easily satisfied. The second is satisfied since the gradient is finite and the bound $\rho$ exists. For the third condition, such learning rates are also easy to be found.
>
> **Q8: "In absence of competing targeted attacks the authors make straightforward modifications to untargeted attacks to get baselines. That this would lead to weak baselines was predictable, and the comparison isn't too fair. Providing it is better than not, but I do not think it is an essential part of the paper, and it could be moved to an appendix without loss."**
>
> > **A8**: These baselines are the strongest adversarial attack methods in RL. For fair comparison, we use the same method to learn a reward function. Figure 3 and 4 demonstrate the effectiveness of PALM and even the strongest untargeted attack methods cannot work. We even provide baseline methods with ground-truth reward function and the results show that PALM exceeds the oracle in most experiments.
>
> **Reference**
>
> [1] Zhang et al. Robust reinforcement learning on state observations with learned optimal adversary. ICLR 2021.
>
> [2] Sun et al. Who is the strongest enemy? Towards optimal and efficient evasion attacks in deep RL. ICLR 2022.
>
> [3] Goodfellow et al. Explaining and harnessing adversarial examples. ICLR 2015.
>
> [4] Zhang et al. Attack on practical speaker verification system using universal adversarial perturbations. ICASSP 2021.
>
> [5] Eykholt et al. Robust physical-world attacks on deep learning visual classification. CVPR 2018.
>
> [6] Pattanaik et al. Robust deep reinforcement learning with adversarial attacks. AAMAS 2018.
>
> [7] Huang et al. Adversarial attacks on neural network policies. arXiv 2017.
>
> [8] Zhang et al. Robust deep reinforcement learning against adversarial perturbations on observations. NeurIPS 2020.
>
> [9] Kingma et al. Adam: A method for stochastic optimization. ICLR 2015.

---

> > ### Comment · Reviewer_2Rf7 · 2022-11-15
> > **Followup question**
> >
> > I will reformulate questions Q4, Q5 and Q6. By "threat model" I do not particularly wonder about $l_\infty$ norm, but rather your general framework for perturbing the environment. My confusion stemmed from this excerpt:
> > *Specifically, the adversary πα perturbs the state s into  ̃s which is restricted by B(s) (i.e.,  ̃s ∈ B(s)); B(s) is defined as a set { ̃s ∈ S :∥s −  ̃s ∥p≤ ε}, which limits the attack power of the adversary. The victim takes action according to the observed  ̃s and true states in the environment are not changed*
> >
> > However, I believe I understand the setting now. It seems that by "true states in the environment are not changed" you mean that the correct policy is the one that would occur the original environment. However the environment has (imperceptibly) changed. So I find that sentence slightly confusing and needing clarification.
> >
> > I think that Figure 1 contributes to that confusion because you do not show an example of perturbed state and jump to the outcome (the modified action). Among your references above for instance, Figure 1 in [7] gives a good example of the kind of illustration that would strengthen your paper.
> >
> > I appreciate the rest of the clarifications in your revision.

---

> > > ### Author Response · Authors · 2022-11-17
> > > **Response to Reviewer 2Rf7**
> > >
> > > We thank Reviewer 2Rf7 for the response and provide our point-wise response below.
> > >
> > > **Q1: "I will reformulate questions Q4, Q5 and Q6. By "threat model" I do not particularly wonder about $\ell_\infty$ norm, but rather your general framework for perturbing the environment."**
> > >
> > > > **A1**: We aim to manipulate the well-performed DRL agents without getting access to the parameters of victim and only query the victim's actions. The full procedure of targeted attack is as follows. The adversary first receives the true state $s$ from the environment and perturbs it into $\tilde{s}$. Specifically, the adversary produces a Gaussian noise $\Delta$ and generates the perturbed state through $\tilde{s} = s+\Delta*\epsilon$. Then, the victim observes $\tilde{s}$ and takes action according to it.
> > >
> > > **Q2: "It seems that by "true states in the environment are not changed" you mean that the correct policy is the one that would occur the original environment. However, the environment has (imperceptibly) changed. So I find that sentence slightly confusing and needing clarification."**
> > >
> > > > **A2**: During manipulation, the true environment is not changed, and PALM only perturbs the observations of the victim. The victim takes action according to its observations. Therefore, the targeted generation of perturbed states can manipulate the victim to behave as humans desire.
> > >
> > > **Q3: "I think that Figure 1 contributes to that confusion because you do not show an example of perturbed state and jump to the outcome (the modified action). Among your references above for instance, Figure 1 in [7] gives a good example of the kind of illustration that would strengthen your paper."**
> > >
> > > > **A3**: Thanks for your suggestion. We will revise Figure 1 and add a visualization of the perturbed state to it.

---

> > > ### Author Response · Authors · 2022-11-18
> > > **A Reminder to Reviewer 2Rf7**
> > >
> > > Dear Reviewer 2Rf7,
> > >
> > > We have updated the manuscript and supplementary.

---

> > > > ### Comment · Reviewer_2Rf7 · 2022-11-25
> > > > **response to authors**
> > > >
> > > > Thank you. I think the changes brought to the paper (in both revisions) are a step in the right direction. It is still perfectible though, and I retain my opinion that the paper is worth accepting but has potential for significant improvement in form.

---

> ### Author Response · Authors · 2022-11-13
> **Response to Reviewer 2Rf7 (Part 1/2)**
>
> We thank reviewer 2Rf7 for your positive support. We will provide point-wise responses to your questions.
>
> **Q1: "One weak point is clarity ... I am not sure whether the intention policy is learned before, or jointly to the weighing function and adversarial policy (Figure 2 says jointly, but the method seem to imply before)."**
>
> > **A1**: We apologize for the confusion. The intention policy is jointly learned with the adversarial policy and the weighing function. Please refer to Algorithm 1 in Appendix A for details. We will carefully revise our paper to improve clarity.
>
> **Q2: "Another point I am not sure about is the attack budget: it is mentioned in the appendix, and the introduction says that perturbations are imperceptible, but the actual method does not make clear how that budget is handled or why it is necessary. "**
>
> > **A2**: The attack budget limits the adversary's power, and imperceptible perturbation means the observed environment slightly differs from true environment [1]. Specifically, the adversarial policy $\pi_\alpha$ perturbs the state $s$ into $\tilde{s} \in$ {$\tilde{s} | \parallel s-\tilde{s} \parallel_\infty \le \epsilon$}, where $\epsilon$ is the attack budget. The attack budget is bounded by $\ell_\infty$ norm following previous methods [1,2].
>
> **Q3: "Overall I'd suggest having a full part dedicated to the outline of the method, keeping technical aspects to a minimum, and moving the rest (theoretical results, learning rates, etc) somewhere else. More detailed diagrams for subparts of the pipeline would also be useful."**
>
> > **A3**: Thanks for your suggestion and we will rearrange the contents and clarify our method.
>
> **Q4: "Another issue is the limited amount of motivation the authors provide for their work. One sentence in the introduction says that studying adversarial attacks is "crucial", but why some threat models rather than others?**
>
> > **A4**: For fair comparison, we just use $\ell_\infty$ to restrict the budget as previous methods [1,2]. We conduct additional experiments to compare different threat models. The results are shown in the following table. We notice that PALM works fail on $\ell_1$ and $\ell_2$ treat model. In these models, perturbation apportions on each dimension are tiny due to a finite attack budget, leading to poor performance. Meanwhile, plenty of researches [1,2,3,4] focus on $\ell_\infty$, which means it is representative. Extending to other threat models is also an interesting direction for furture work.
> >
> > |              | $\ell_1$ | $\ell_2$ | $\ell_\infty$ |
> > | :----------: | :------: | :------: | :-----------: |
> > | Drawer Open  | 1.0%     | 0.0%     | 94.7%         |
> > | Faucet Close | 6.0%     | 1.0%     | 96.8%         |
>
> **Q5: "With an attack like PALM what could an attacker concretely do in the real world? Motivating or defining informally specific aspects of the work could be useful as well."**
>
> > **A5**: Using adversarial attack methods in the real world needs some extra constraints. For example, to attack autonomous vehicles to move forward when observing a "STOP" [5], the perturbations should be able to be printed by a printer. And extending PALM to real world also needs some constraints, such as PALM can obtain the true states of the victim and secretly perturb victim's observation. We consider real world attacks against DRL agents as a future work.
>
> **Q6: "What is an "imperceptible perturbation"? What imperceptible means on image or speech is clear, but on RL states it is harder to grasp."**
>
> > **A6**: Thank you for your question. In RL, imperceptible perturbation means the observed environment perturbed by the adversary slightly differs from the true environment [1]. In our method, we consider $L^p$ attack ($\parallel s-\tilde{s} \parallel_p \le \epsilon$), which is a common threat model in adversarial RL [1,6,7,8] and the perturbation is restricted by the budget $\epsilon$. Under the limitation of the budget, the perturbation is too small to be detected by the agent.
>
> (Citations included in Part 2)

---

### Official Review · Reviewer_nMXe · 2022-11-03

**Confidence:** 2
**Correctness:** 3
**Technical Novelty And Significance:** 3
**Empirical Novelty And Significance:** 3
**Recommendation:** 6

**Clarity, Quality, Novelty And Reproducibility:**

- Novelty: This work presents a novel idea of targeted attack against DRL agents by leveraging preference-based RL.
- Clarity: This work is kind of hard to follow. Moreover, Section 4.3 provides convergence guarantees for the proposed method. But I don’t understand how this section helps explain the efficiency or optimality of the proposed method. If the intention policy is not optimal, could the adversarial policy still be optimal? Especially in Equation 7, if \theta is not optimal, even if the \alpha is the argmin of L_att, does it still imply the optimality of the attack?
Quality: This work has some minor issues in terms of the presentation.
Reproducibility: The author provides a link to the code and some demo. I trust the authors regarding the experimental results.


**Strength And Weaknesses:**

#### **Strengths**:
- This work focuses on a novel targeted attack setting.
#### **Weaknesses**:
- In general, I found this work kind of hard to follow given the complicated notations. Based on my understanding, PALM uses RL to attack an RL agent.
- Moreover, I am confused by Equation 4. What is the target soft Q-function? If we already know the targeted Q-function \bar{\phi}, why don’t we just use it in Equation 3 directly? If we don’t know \bar{\phi}, how can we optimize \phi with Equation 4?
- On page two, the authors mentioned several works that could improve the robustness of policies. Although I agree that this paper focuses on designing attack instead of defense, I am still wondering whether the proposed attack could surpass such defenses. Could the authors justify or provide some discussions about whether the proposed method could break these defenses or why they didn't run experiments on this? What if the proposed attack could not work under a even simple defense?
- I also have a question about Section 4.3. (please see below in clarity)


**Summary Of The Paper:**

This paper presents a novel targeted preference-based adversarial attack against deep reinforcement learning (DRL) agents, so that the DRL agents would show extreme behaviors desired by adversaries. In particular, the proposed PALM adopts human preference, an intention policy and a weighting function to guide the adversary. Moreover, the authors also provided a theoretical analysis for the convergence of PALM. Finally, the experimental results suggest that PALM outperforms other baselines under the targeted attack setting.

**Summary Of The Review:**

Overall, I think this work is interesting. However, as stated above, I didn’t fully understand Equation 4 and its purpose, which might affect my understanding and judgement of this paper and its contribution. Currently, I rate it as 5, but I am willing to increase my score if the authors could answer my questions above. Thank you very much!

---

> ### Author Response · Authors · 2022-11-13
> **Response to Reviewer nMXe**
>
> We thank reviewer nMXe for the constructive comments. We will give our point-wise responses below.
>
> **Q1: "In general, I found this work kind of hard to follow given the complicated notations. Based on my understanding, PALM uses RL to attack an RL agent."**
>
> > **A1**: We apologize for the confusion. Our goal is to train a well-performing adversarial policy which can manipulate the victim policy to behave as human desire. To achieve this, we formulate the problem as a GSA-MDP. It differs from SA-MDP [1] in that it involves a reward function consistent with human preferences. We will add detailed notations and descriptions in the revised version.
>
> **Q2: "I am confused by Equation 4. What is the target soft Q-function? If we already know the targeted Q-function $\bar{\phi}$, why don’t we just use it in Equation 3 directly? If we don’t know $\bar{\phi}$, how can we optimize $\phi$ with Equation 4?"**
>
> > **A2**: The target Q-function is the same neural network as Q-function, whose parameters $\bar{\phi}$ are only updated with the Q-network parameters $\phi$ every C step. It is a common trick in RL for stable training [2,3]. We will add more details about SAC [3] to avoid confusion.
>
> **Q3: "On page two, ... Could the authors justify or provide some discussions about whether the proposed method could break these defenses or why they didn't run experiments on this? What if the proposed attack could not work under a even simple defense?"**
>
> > **A3**: Thank you for your question. We additionally conduct experiments to evaluate the robustness of adversarially trained agents. According to the implementation of [1,4], We respectively train robust policies (SA-ATLA, PA-ATLA) under the attacks of SA-RL and PA-AD and use PALM to attack them. The results in the following table show that PALM can break these defences.
> >
> > |     Task     |  Model  | Natural Reward | PALM         |
> > | :----------: | :-----: | :------------: | :----------: |
> > | Drawer Open  | SA-ATLA | 4705$\pm$36    | 378$\pm$11   |
> > | Drawer Open  | PA-ATLA | 4710$\pm$18    | 422$\pm$183  |
> > | Faucet Open  | SA-ATLA | 4720$\pm$6     | 610$\pm$293  |
> > | Faucet Open  | PA-ATLA | 4721$\pm$5     | 438$\pm$53   |
> > | Faucet Close | SA-ATLA | 4735$\pm$14    | 1362$\pm$149 |
> > | Faucet Close | PA-ATLA | 4690$\pm$120   | 661$\pm$279  |
>
> **Q4: "Section 4.3 provides convergence... If the intention policy is not optimal, could the adversarial policy still be optimal? Especially in Equation 7, if $\theta$ is not optimal, even if the $\alpha$ is the argmin of $L_{att}$, does it still imply the optimality of the attack?"**
>
> > **A4**: Thank you for your question. The performance of intention policy depends on the amount of human preferences, which means few preference labels may lead to suboptimality. As the results in Table 1 show, a suboptimal intention policy negatively impacts on the performance of the adversarial policy.
>
> **Reference**
>
> [1] Zhang et al. Robust reinforcement learning on state observations with learned optimal adversary. ICLR 2021.
>
> [2] Mnih et al. Human-level control through deep reinforcement learning. Nature 2015.
>
> [3] Haarnoja et al. Soft actor-critic: Off-policy maximum entropy deep reinforcement learning with a stochastic actor. ICML 2018.
>
> [4] Sun et al. Who is the strongest enemy? Towards optimal and efficient evasion attacks in deep RL. ICLR 2022.

---

> > ### Author Response · Authors · 2022-11-17
> > **A gentle reminder**
> >
> > Dear Reviewer nMXe,
> >
> > We would like to know whether we have addressed your concerns. If you have any further questions, we are happy to discuss.

---

> > > ### Comment · Reviewer_nMXe · 2022-11-17
> > > **Response to the authors**
> > >
> > > Dear Authors,
> > >
> > > thank you very much for making the explanations to me. I think now I agree with the formulation of Equation 4, and I agree with the contribution of this work. I will update my grade to support its acceptance.

---

### Author Response · Authors · 2022-11-14
**General response**

We thank reviewers for the constructive comments. We have updated the manuscript and supplementary and the changes are highlighted in blue color. The main changes are:
1) We carefully revised Section 3 and 4 to make our method clear.
2) We added a detailed review of targeted attacks against DRL agents. (Reviewer kjbK)
3) We added Figure 3 to provide a diagram of preference-based RL. (Reviewer 2Rf7)
4) We added Figure 7 to demonstrate the increased exploration ability by the intention policy. (Reviewer kjbK, x6Lt)

---

### Decision · Program_Chairs · 2023-01-20

**Decision:**

Reject

**Justification For Why Not Higher Score:**

The reviewers shared a common concern that several claims made in the paper lack strong support and reached a rejection decision.

**Justification For Why Not Lower Score:**

N/A

**Metareview: Summary, Strengths And Weaknesses:**

The reviewers agreed that the paper considers an interesting setting for targeted adversarial attacks against deep RL agents and acknowledged the novelty of using a preference-based RL framework as an attack method. However, the reviewers pointed out several weaknesses and shared a common concern that several claims made in the paper lack strong support (in terms of experiments or theory). We want to thank the authors for their detailed responses; the paper was also discussed among all the reviewers, considering the responses and revision. Based on the raised concerns and follow-up discussions, unfortunately, the final decision is a rejection. Nevertheless, the reviewers have provided detailed and constructive feedback. We hope the authors can incorporate this feedback when preparing future revisions of the paper.